# Chain-of-Goals Hierarchical Policy for
# Long-Horizon Offline Goal-Conditioned RL

Jinwoo Choi [1]   Sang-Hyun Lee [2] *   Seung-Woo Seo [1] *

## Abstract

Offline goal-conditioned reinforcement learning remains challenging for long-horizon tasks. While hierarchical approaches mitigate this issue by decomposing tasks, most existing methods rely on separate high- and low-level networks and generate only a single intermediate subgoal, leaving several structural limitations in long-horizon decision-making. To address this limitation, we draw inspiration from chain-of-thought reasoning and propose the Chain-of-Goals Hierarchical Policy (CoGHP), a novel framework that reformulates hierarchical decision-making as autoregressive sequence modeling within a unified architecture. Given a state and a final goal, CoGHP autoregressively generates a sequence of latent subgoals followed by the primitive action, where each latent subgoal acts as a reasoning step that conditions subsequent predictions. To implement this efficiently, we introduce an MLP-Mixer backbone, which supports cross-token communication and captures structural relationships among state, goal, latent subgoals, and action. Across challenging navigation and manipulation benchmarks, CoGHP consistently outperforms strong offline baselines, demonstrating improved performance on long-horizon tasks. Project page: https://wlsdn9350.github.io/projects/coghp/

## 1. Introduction

Offline goal-conditioned reinforcement learning (RL) (Chebotar et al., 2021; Yang et al., 2022; Ma et al., 2022)

* Equal Advising.  [1]Department of Electrical and Computer Engineering, Seoul National University, Seoul, South Korea [2]Department of Automotive Engineering, Ajou University, Gyeonggi-do, South Korea. Correspondence to: Seung-Woo Seo <sseo@snu.ac.kr>, Sang-Hyun Lee <sanghyunlee@ajou.ac.kr>.

*Proceedings of the 43rd International Conference on Machine Learning*, Seoul, South Korea. PMLR 306, 2026. Copyright 2026 by the author(s).

aims to learn a policy that reaches specified goals using only static, pre-collected datasets, which is useful when interactions with environments are costly or unsafe. However, as horizons expand, the gap between optimal and sub-optimal action values diminishes due to discounting and compounded Bellman errors, leading to an unreliable policy (Park et al., 2023). Offline hierarchical RL (Gupta et al., 2019; Park et al., 2023; Schmidt et al., 2024) addresses this by decomposing tasks into high-level subgoal selection and low-level control, but traditional approaches face a fundamental structural limitation. Most existing hierarchical RL methods rely on two-level hierarchical structures with separate networks for high-level and low-level policies. This architectural separation leads to three critical limitations. First, these approaches typically generate only a single intermediate subgoal, making them inadequate for complex tasks that require coordinating multiple intermediate decisions. Second, when the high-level policy generates erroneous subgoals, the low-level policy blindly executes toward these misguided targets. As a result, it loses awareness of the final goal and may select sub-optimal actions. Third, training hierarchy levels under separate objectives prevents end-to-end gradient flow, blocking corrective signals from propagating across decision-making stages and hindering the coordinated multi-stage reasoning necessary for long-horizon tasks.

How can we develop a unified approach that naturally scales to multiple hierarchy levels while maintaining both computational efficiency and learning stability? Rather than adding more separate networks to handle longer horizons, we need a fundamentally different architectural paradigm that can handle multi-step sequences of intermediate decisions within a single, cohesive framework. We find that a compelling answer to this question can be inspired by the sequential decomposition principle underlying the chain-of-thought (CoT) reasoning paradigm (Wei et al., 2022), where complex problems are decomposed into a sequence of intermediate steps before reaching a final conclusion. By adapting this sequential decomposition principle to hierarchical decision-making, we may be able to effectively address the three critical limitations of traditional offline hierarchical RL. First, analogous to how CoT uses intermediate reasoning steps to tackle complex problems, hierarchical policy

**Chain-of-goals Hierarchical Policy (CoGHP)**

*Figure 1.* **Chain-of-Goals Hierarchical Policy (CoGHP).** CoGHP autoregressively generates a sequence of latent subgoals and the primitive action within a unified model. Each subgoal serves as a reasoning token, providing the agent with sufficient guidance to reach the goal. Autoregressive generation ensures that later predictions build upon earlier ones while maintaining awareness of the final goal. To ensure that the subgoal closest to the agent carries the most informative signal, the sequence is generated in reverse order, beginning with the farthest subgoal relative to the current state and progressing toward the nearest one.

architecture can be reformulated to generate a sequence of multiple intermediate subgoals. Second, this structure could preserve awareness of the final goal by maintaining it as a constant condition throughout the sequence generation. Furthermore, by consolidating the hierarchy into a single unified model, we could enable seamless information and training signal flow across all decision-making stages.

Building on this insight, we introduce the **Chain-of-Goals Hierarchical Policy (CoGHP)**, which draws inspiration from the chain-of-thought paradigm to design a novel architecture for offline goal-conditioned RL. Instead of relying on separate networks for different hierarchy levels, CoGHP reformulates hierarchical decision-making as the autoregressive sequential generation of latent subgoals and the primitive action within a unified model (Figure 1). Each latent subgoal functions as a reasoning step that provides intermediate information carried forward to guide subsequent predictions. Autoregressive generation ensures that later predictions build upon earlier ones while preserving access to the final goal. This chain-of-thought-style structure, from input through intermediate reasoning steps to primitive action, has recently emerged as a useful paradigm for robotic control in vision-language-action models (Zawalski et al., 2024; Chen et al., 2025). CoGHP takes a step further by extending this perspective to the offline goal-conditioned RL setting and instantiating it as a unified hierarchical decision-making framework. To effectively realize this sequence modeling, we introduce the MLP-Mixer (Tolstikhin et al., 2021) architecture as a backbone for hierarchical RL. Its simple feedforward design enables efficient cross-token communication, making it well-suited for processing a sequence of state, goal, latent subgoals, and action. Finally, we train this unified architecture with a shared value function learned from offline data, which provides training signals for all sequence elements, including both intermediate subgoals and primitive actions. This training strategy allows gradient-

based error correction to propagate seamlessly across the entire hierarchy.

In summary, our contributions are threefold. First, we introduce a novel framework that adapts the chain-of-thought reasoning paradigm to offline hierarchical RL, reformulating hierarchical decision-making as autoregressive sequence generation of intermediate subgoals that act as reasoning steps. Second, we introduce the MLP-Mixer architecture as a backbone for hierarchical RL, leveraging its efficient cross-token communication to enable unified end-to-end training across all decision-making stages. Third, we demonstrate that CoGHP outperforms strong baselines on challenging navigation and manipulation benchmarks, validating its effectiveness for long-horizon offline control tasks.

## 2. Related Work

**Offline Hierarchical RL**    Prior work in offline RL has primarily tackled distribution shift and overestimation through regularization and constraint-based methods (Kostrikov et al., 2021; Kumar et al., 2020; Wu et al., 2019), but these approaches still struggle on long-horizon tasks. To address this issue, offline hierarchical RL decomposes decision-making into temporally abstract subgoals and low-level control. Key directions include skill and primitive discovery from static data (Ajay et al., 2020; Krishnan et al., 2017; Pertsch et al., 2021; Choi & Seo, 2025; Pertsch et al., 2022), latent plan representation learning for efficient high-dimensional planning (Jiang et al., 2022; Rosete-Beas et al., 2023; Lynch et al., 2020; Shah et al., 2021), integrated hierarchical planners combining subgoal selection with goal-conditioned controllers (Park et al., 2023; Gupta et al., 2019; Schmidt et al., 2024; Li et al., 2022; Ahn et al., 2026; Chen et al., 2024), and model-based world modeling for offline planning (Shi et al., 2022; Freed et al., 2023). Recent diffusion-based planning methods have also been used

to generate, compose, or search over trajectories for long-horizon decision-making (Chen et al., 2024; Luo et al., 2026; Yoon et al., 2025; 2026). Complementary to these prior directions, CoGHP reformulates hierarchical decision-making as a unified autoregressive sequence modeling problem, producing multiple subgoals and primitive actions within a single architecture that enables end-to-end optimization.

**Chain-of-Thought** Chain-of-thought prompting was first shown to unlock complex reasoning in large language models by eliciting intermediate rationales, yielding dramatic gains on arithmetic, commonsense, and symbolic benchmarks (Wei et al., 2022). Subsequent work refined its application through analysis of prompting factors and enhanced reasoning via automatic rationale synthesis, self-consistency decoding, and progressive problem decomposition (Sprague et al., 2024; Zhang et al., 2022; Wang et al., 2022a; Zhou et al., 2022). In robotics and embodied AI, chain-of-thought-inspired intermediate planning has been applied to vision-language agents, navigation, policy learning with semantic subgoals, sensorimotor grounding, affordance-based action planning, and action-level autoregressive trajectory generation for manipulation (Mu et al., 2023; Lin et al., 2025; Zawalski et al., 2024; Brohan et al., 2023; Zhang et al., 2026). Building on this foundation, we propose to bring chain-of-thought-inspired sequential decomposition into offline goal-conditioned RL, in which latent subgoals serve as reasoning steps that carry forward intermediate context for subsequent predictions.

**MLP-Mixer** MLP-Mixer introduced a minimalist all-MLP backbone for vision by alternately applying token-mixing and channel-mixing MLPs to patch embeddings, achieving competitive classification performance without convolutions or attention (Tolstikhin et al., 2021). Subsequent extensions have applied the same principles beyond standard imaging: dynamic token mixing for adaptive vision models (Wang et al., 2022b), and fully MLP-based architectures for multivariate time series forecasting (Chen et al., 2023; Cho & Lee, 2025; Wang et al., 2024). These advances underscore the MLP-Mixer's linear scaling and representational flexibility across modalities. To the best of our knowledge, CoGHP is the first work to adapt MLP-Mixer for offline goal-conditioned RL, enabling unified autoregressive sequence generation of hierarchical subgoals within a single end-to-end framework.

## 3. Problem Formulation and Preliminaries

### 3.1. Problem Formulation

We study offline goal-conditioned reinforcement learning in a Markov decision process $M = (\mathcal{S}, \mathcal{A}, P, r, \gamma, \rho_0)$, where $\mathcal{S}$ is the state space, $\mathcal{A}$ denotes the action space, $P(s' \mid s, a)$ denotes the transition dynamics, $r(s, g)$ denotes a reward

function measuring progress toward goal $g$, $\gamma \in [0, 1)$ denotes the discount factor, and $\rho_0$ denotes the initial state distribution. A static dataset $\mathcal{D} = \{\tau_i\}_{i=1}^N$ of trajectories $\tau = (s_0, a_0, s_1, a_1, \dots, s_T)$ is collected beforehand, and no further environment interaction is permitted. We assume a goal space $\mathcal{G} = \mathcal{S}$, and at evaluation time, each episode is paired with a goal $g \sim p(g)$. The objective is to learn a stationary policy $\pi_\theta(a \mid s, g)$ that maximizes the expected discounted return $J(\pi_\theta) = \mathbb{E}_{g \sim p(g), \tau \sim \pi_\theta}[\sum_{t=0}^T \gamma^t r(s_t, g)]$.

### 3.2. Goal-conditioned Implicit Q-Learning (IQL)

Implicit Q-Learning (IQL) (Kostrikov et al., 2021) stabilizes offline RL by avoiding queries to out-of-distribution (OOD) actions through two key components: a state-value function $V_\psi(s)$ and an action-value function $Q_\theta(s, a)$. The value functions are trained via:

$$\mathcal{L}_Q(\theta) = \mathbb{E}_{(s,a,s') \sim \mathcal{D}} \left[ \left( r(s,a) + \gamma V_\psi(s') - Q_\theta(s,a) \right)^2 \right], \tag{1}$$

$$\mathcal{L}_V(\psi) = \mathbb{E}_{(s,a) \sim \mathcal{D}} \left[ L_2^\tau \left( Q_{\bar{\theta}}(s,a) - V_\psi(s) \right) \right], \tag{2}$$

where $L_2^\tau(x) = |\tau - \mathbb{1}(x < 0)|x^2$ and $\tau \in [0.5, 1)$ controls conservatism (higher $\tau$ prioritizes optimistic returns), and $\bar{\theta}$ are the parameters of the target Q network. The policy $\pi_\phi(a|s)$ is then extracted via advantage-weighted regression (AWR) (Peters & Schaal, 2007; Wang et al., 2020):

$$J_\pi(\phi) = \mathbb{E}_{(s,a) \sim \mathcal{D}} \left[ \exp \left( \beta \cdot A(s,a) \right) \log \pi_\phi(a|s) \right], \tag{3}$$

with $A(s,a) = Q_\theta(s,a) - V_\psi(s)$, and $\beta$ is the inverse temperature parameter.

For goal-conditioned RL, IQL is extended to learn a goal-conditioned state-value function $V_\psi(s, g)$, preserving IQL's key advantage of stable value learning without requiring explicit Q-function evaluations on out-of-distribution actions (Ghosh et al., 2023):

$$\mathcal{L}_V(\psi) = \mathbb{E}_{(s,s',g) \sim \mathcal{D}}[L_2^\tau(r(s,g) + \gamma V_{\bar{\psi}}(s',g) - V_\psi(s,g))]. \tag{4}$$

The corresponding policy is trained via a variant of AWR, which reweights behavior actions by exponentiated estimates of the goal-conditioned advantage:

$$J_\pi(\phi) = \mathbb{E}_{\substack{(s,a,s') \sim \mathcal{D}, \\ g \sim p(g)}}[\exp(\beta \cdot A(s,a,g)) \log \pi_\phi(a|s,g)], \tag{5}$$

where $A(s, a, g) \approx \gamma V_\psi(s', g) + r(s, g) - V_\psi(s, g)$. This advantage-weighted policy extraction ensures that the learned policy focuses on high-value actions relative to each specific goal.

### 3.3. MLP-Mixer

MLP-Mixer (Tolstikhin et al., 2021) is a simple, all-MLP architecture that was originally introduced for image classification tasks such as ImageNet and CIFAR. It avoids both convolution and self-attention, instead relying on alternating multi-layer perceptron blocks over spatial and channel dimensions to achieve competitive visual recognition performance using only MLPs. In its implementation, an input image is first divided into fixed-size patches and linearly projected to a sequence of token embeddings. Each Mixer layer then interleaves two MLP sub-layers: a token-mixing MLP that operates across the patch dimension to exchange information between spatial locations, and a channel-mixing MLP that acts independently on each token's feature channels to capture per-location feature interactions. Both sublayers are wrapped with layer normalization, residual connections, and pointwise nonlinearities (e.g., GELU).

## 4. Proposed Method

We present the **Chain-of-Goals Hierarchical Policy (CoGHP)**, a novel framework that brings chain-of-thought reasoning to offline goal-conditioned RL. Our proposed approach addresses the fundamental limitations that plague most existing offline hierarchical RL methods: single subgoal constraints, loss of final goal awareness when high-level guidance is erroneous, and fragmented optimization across separate networks. Our key insight is to reformulate hierarchical decision-making as a sequence generation problem, where the policy autoregressively generates a sequence of latent subgoals and the primitive action, all conditioned on both the current state and the goal state. This formulation preserves final goal awareness and enables end-to-end optimization across all decision stages. This section details our architectural design (Section 4.1), describes the sequence generation mechanism (Section 4.2), presents the training objectives (Section 4.3), and outlines the training procedure (Section 4.4).

### 4.1. Architecture Design

To implement this sequence generation paradigm, we require an architecture that can efficiently process a sequence of embedded tokens (state, goal, latent subgoals, and action) while modeling dependencies between sequence elements. For such sequential processing requirements, Transformer architectures are widely adopted across language and vision domains. However, while Transformers excel at capturing complex inter-element dependencies and dynamic

interactions, they are less suited for settings where tokens have fixed position-dependent roles and the target signal primarily depends on its own temporal position rather than complex interactions (Zeng et al., 2023; Chen et al., 2023). Consequently, within our offline hierarchical RL framework, where each token position is assigned a fixed semantic role (such as current state, final goal, latent subgoal sequence, and primitive action), we found Transformer backbones to offer limited generalization benefits and to exhibit reduced training stability in practice. This observation is empirically supported by our ablation results.

Our key architectural insight is to harness the MLP-Mixer architecture, which proves well-suited for sequence generation with position-dependent token roles. MLP-Mixer consists of alternating token-mixing and channel-mixing MLP layers that enable cross-token communication and per-token feature refinement using only simple feedforward operations. While MLP-Mixer is inherently sensitive to input token order and does not require separate positional embeddings, we augment it with a learnable causal token-mixer to better incorporate information from previously generated tokens during autoregressive subgoal and action generation. The causal mixer is implemented as a lower-triangular matrix applied to the stacked tokens, transforming each token into a weighted sum of previously generated tokens. This design enables better incorporation of sequential dependencies crucial for hierarchical decision-making. Detailed specifications are provided in Appendix A.1.

### 4.2. Sequence Generation

Our hierarchical policy first takes as input the token sequence $[e_o, e_g, \tilde{z}_H, \ldots, \tilde{z}_1, \tilde{z}_a]$, where $e_o = \phi_o(s)$ and $e_g = \phi_g(g)$ are state and goal embeddings, $\tilde{z}_{1:H}$ are learnable subgoal initial tokens, and $\tilde{z}_a$ is the action initial token. These initial tokens are progressively filled in an autoregressive manner. At generation step $i$, the policy takes as input the state, the goal, all previously generated subgoals $z_{i+1:H}$, and the remaining initial tokens $\tilde{z}_{1:i}, \tilde{z}_a$. Importantly, we hypothesize that subgoals closer to the current state should incorporate more comprehensive information from the hierarchical reasoning process. Therefore, we configure our model to generate subgoals sequentially from those most distant from the current state ($z_H$) to those nearest ($z_1$), a choice empirically supported by our analysis in Appendix C.3. The MLP-Mixer backbone processes this sequence through token-mixer, causal mixer, and channel-mixer layers to output hidden state $h_i$, which is then passed through subgoal head $\phi_s$ to produce $z_i = \phi_s(h_i)$. After all $H$ latent subgoals are generated, the hidden state $h_a$ derived from the action initial token $\tilde{z}_a$ is passed through the action head $\phi_a$ to produce the primitive action $a = \phi_a(h_a)$. The complete sequence generation can be written as:

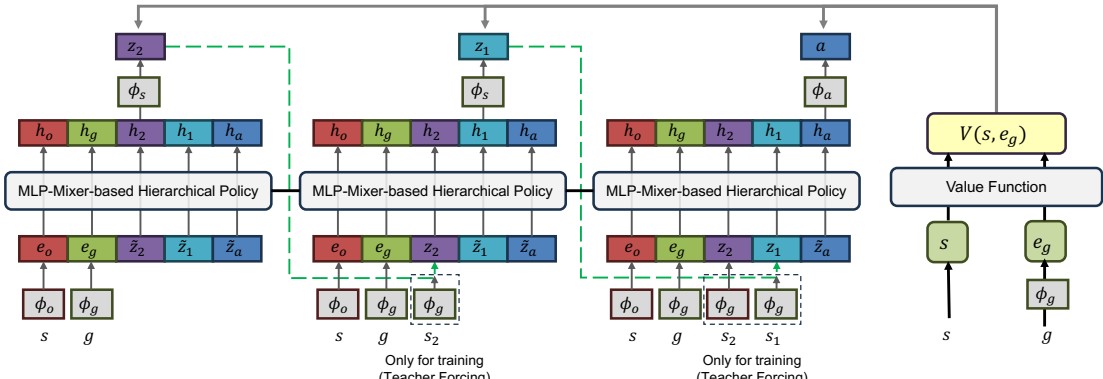

*Figure 2.* **Autoregressive Sequence Generation in CoGHP.** The policy autoregressively generates latent subgoals in order from most distant ($z_H$) to nearest ($z_1$) from the current state, and the primitive action $a$. At step $i$, the MLP-Mixer processes state embedding $e_o$, goal embedding $e_g$, previously generated subgoals, and remaining initial tokens to output $z_i$. This sequential generation ensures that each subgoal leverages information from all previously generated waypoints, enabling comprehensive hierarchical reasoning. During training, we apply teacher forcing by providing ground-truth subgoal embeddings to prevent error accumulation.

$$\pi_\theta(z_{1:H}, a \mid s, g) = \left( \prod_{i=H}^{1} \pi_\theta(z_i \mid e_o, e_g, z_{i+1:H}, \tilde{z}_{1:i}, \tilde{z}_a) \right) \cdot \pi_\theta(a \mid e_o, e_g, z_{1:H}, \tilde{z}_a), \quad (6)$$

where the product notation $\prod_{i=H}^{1}$ indicates the generation order from $z_H$ to $z_1$, and $z_{i+1:H} = \emptyset$ when $i = H$. This autoregressive sequence generation mechanism is visualized in Figure 2.

## 4.3. Training Objectives

To train this unified architecture, we propose an AWR-style objective that provides consistent training signals across all sequence elements. Our approach employs a shared value function to train both the latent subgoal sequence generation and final action prediction within the same network, ensuring coherent optimization across all hierarchical levels. While this shared value-based training strategy draws inspiration from HIQL (Park et al., 2023), our key innovation lies in unifying all hierarchy levels within a single network architecture, contrasting with HIQL's approach of using separate network modules for different hierarchical components.

First, we learn $V_\psi(s, e_g)$ from the offline dataset $\mathcal{D}$ by minimizing the IQL temporal-difference error as defined in Equation 4. By training the value function on state-embedded goals, we can directly apply it to both embedded goals and latent subgoals, which reside in the same latent space. For details on value function training, please refer to Appendix A.2. Using advantage estimates derived from this value function, we define separate objectives for each prediction step. Our training objectives are:

$$J^{h_i}(\theta) = \mathbb{E}_{(s,g,s_{i:H}) \sim \mathcal{D}} \Big[ \exp(\beta \cdot \tilde{A}^h(s, s_i, e_g)) \\ \cdot \log \pi_\theta(z_i | e_o, e_g, z_{i+1:H}) \Big], \quad (7)$$

$$J^\ell(\theta) = \mathbb{E}_{(s,a,s',s_{1:H},g) \sim \mathcal{D}} \Big[ \exp(\beta \cdot \tilde{A}^\ell(s, a, z_1)) \\ \cdot \log \pi_\theta(a | e_o, e_g, z_{1:H}) \Big], \quad (8)$$

where $\beta$ is a temperature parameter controlling the sharpness of advantage weighting. For notational simplicity, we omit the remaining initial tokens $\tilde{z}_{1:i}$ and $\tilde{z}_a$ from the conditioning of $\pi_\theta(\cdot)$. Following HIQL's advantage approximations, we use $\tilde{A}^h(s, s_i, e_g) \approx V_\psi(s_i, e_g) - V_\psi(s, e_g)$ and $\tilde{A}^\ell(s, a, z_1) \approx V_\psi(s', z_1) - V_\psi(s, z_1)$. The advantage terms quantify the value of each prediction step, where $\tilde{A}^h(s, s_i, e_g)$ measures the benefit of reaching intermediate state $s_i$ toward goal $g$, and $\tilde{A}^\ell(s, a, z_1)$ evaluates action quality relative to the nearest generated subgoal. We extract target subgoals $s_{1:H}$ by sampling states at fixed $k$-step intervals along dataset trajectories, providing supervision for our latent subgoals $z_{1:H}$ to learn meaningful waypoint representations. This advantage weighting naturally guides the policy toward high-value subgoals and corresponding optimal actions.

These individual objectives are aggregated into a single end-to-end loss:

$$J_{\text{total}}(\theta) = \lambda_h \sum_{i=H}^{1} \gamma_h^{i-1} J^{h_i}(\theta) + \lambda_\ell J^\ell(\theta), \quad (9)$$

where $\lambda_h$ and $\lambda_\ell$ are weight coefficients for the subgoal and action losses, respectively, and $\gamma_h$ is the discount factor

that down-weights contributions from distant subgoals. The summation notation $\sum_{i=H}^{1}$ denotes the computation order from $J^{h_H}$ to $J^{h_1}$.

### 4.4. Training Procedure

CoGHP employs an alternating optimization scheme between the value function and the hierarchical policy. In each iteration, we first update $V_\psi$ using sampled transitions $(s, s', g)$, then fix the value function and train the policy $\pi_\theta$ using trajectory segments $(s, a, s', s_{1:H}, g)$. During policy training, we apply teacher forcing by providing ground-truth subgoal embeddings instead of using the policy's own predictions, preventing error accumulation during early training stages. Further details can be found in Appendix A.3.

For simplicity, we implement the generated latent subgoals as encoded future states. While the MLP-Mixer-based backbone of CoGHP can, in principle, support alternative subgoal representations (e.g., learned skill primitives (Pertsch et al., 2021) or abstract semantic embeddings (Brohan et al., 2023)), doing so would likely require additional data modalities or annotations, along with dedicated training objectives. We leave these extensions to future work.

## 5. Experiments

We conduct experiments to evaluate our approach through three sets of evaluations. First, we evaluate CoGHP's performance against strong baselines on challenging navigation and manipulation tasks. Second, we analyze the contribution of our architectural components through ablation studies by comparing CoGHP to a Transformer-based variant and to an MLP-Mixer variant without the causal mixer, showing that CoGHP becomes increasingly advantageous as task complexity grows. Third, we visualize the latent subgoal sequences generated by CoGHP to provide insights into how our chain-of-goals approach decomposes complex tasks.

### 5.1. Experimental Setup

We evaluate CoGHP on the OGBench suite (Park et al., 2024), a comprehensive benchmark for offline goal-conditioned RL that features diverse navigation and manipulation environments (Figure 3). The navigation tasks include pointmaze, which involves controlling a 2-D point mass, and antmaze, which involves controlling a quadrupedal Ant agent with 8 degrees of freedom. We test across three environment sizes (medium, large, and giant) to assess how CoGHP's long-horizon reasoning capabilities scale with increasing maze complexity. For manipulation tasks, we focus on the cube and scene environments to evaluate distinct aspects of object interaction. The cube task requires arranging blocks into target configurations through pick-and-place operations. We examine single, double, and triple

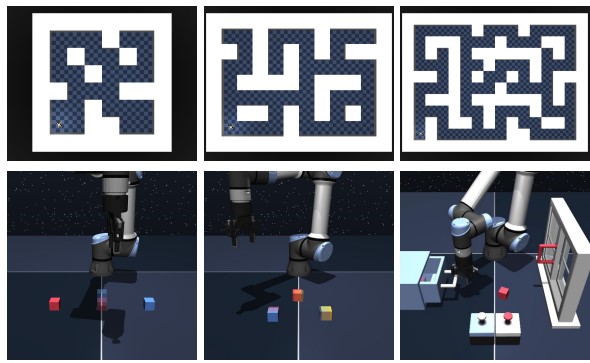

*Figure 3.* **Evaluation Environments.** Our experiments utilize environments from the OGBench suite. **Top row**: Navigation capabilities are tested in maze-medium, maze-large, and maze-giant (left to right) with increasing complexity using both Point mass and Ant agents. **Bottom row**: Manipulation skills are evaluated in cube variants (left and center) involving single to triple cube tasks, and the scene environment (rightmost), which requires multi-step interaction sequences such as unlocking, opening, and manipulating objects.

cube variants to understand how performance scales with the number of objects requiring coordination. The scene environment presents a more sophisticated challenge, demanding multi-step sequential interactions such as unlocking, opening, placing, and closing operations in the correct order. This environment is particularly well-suited for evaluating long-horizon sequential reasoning and the agent's ability to handle diverse, structured object interactions. We additionally report results on the pixel-based tasks with high-dimensional observations (visual-antmaze and visual-cube) in Appendix C.1. Following OGBench's evaluation protocol, we test five predefined state-goal pairs per environment and report the average success rate for all tasks.

We evaluate CoGHP against eight representative algorithms from OGBench and recent offline GCRL studies. Goal-Conditioned Behavioral Cloning (GCBC) (Lynch et al., 2020; Ghosh et al., 2019) formulates goal-conditioned control as a supervised learning problem by cloning the demonstrated action at each state–goal pair. Goal-Conditioned Implicit V-Learning (GCIVL) and Implicit Q-Learning (GCIQL) (Kostrikov et al., 2021) both fit expectile-based value (and Q-value) estimators on offline data and then extract policies through advantage-weighted regression or behavior-constrained actor updates. Quasimetric RL (QRL) (Wang et al., 2023) learns a goal-conditioned value function parameterized as an asymmetric quasimetric, enforcing the triangle inequality as a structural constraint. Contrastive RL (CRL) (Eysenbach et al., 2022) uses a contrastive objective to train a Monte Carlo–style value estimator and performs a single-step policy improvement. Finally, Hierarchical Implicit Q-Learning (HIQL) (Park et al., 2023) leverages a unified value function to derive separate high-level subgoal and low-level action policies via distinct advantage-weighted

*Table 1.* **Experimental Results on Navigation and Manipulation Environments.** Bold values indicate the highest performance values within a 5-point range of the maximum in each row. Standard deviations across 8 random seeds are shown. CoGHP achieves consistently superior or competitive results across diverse environments.

| Env | Dataset | GCBC | GCIVL | GCIQL | QRL | CRL | HIQL | OTA | SAW | CoGHP (ours) |
|---|---|---|---|---|---|---|---|---|---|---|
| **pointmaze** | pointmaze-medium-navigate-v0 | $9 \pm 6$ | $63 \pm 6$ | $53 \pm 8$ | $82 \pm 5$ | $29 \pm 7$ | $79 \pm 5$ | $86 \pm 2$ | $\mathbf{97} \pm 2$ | $\mathbf{99} \pm 1$ |
| | pointmaze-large-navigate-v0 | $29 \pm 6$ | $45 \pm 5$ | $34 \pm 3$ | $86 \pm 9$ | $39 \pm 7$ | $58 \pm 5$ | $85 \pm 5$ | $85 \pm 10$ | $\mathbf{91} \pm 8$ |
| | pointmaze-giant-navigate-v0 | $1 \pm 2$ | $0 \pm 0$ | $0 \pm 0$ | $68 \pm 7$ | $27 \pm 10$ | $46 \pm 9$ | $72 \pm 6$ | $68 \pm 8$ | $\mathbf{79} \pm 8$ |
| **antmaze** | antmaze-medium-navigate-v0 | $29 \pm 4$ | $72 \pm 8$ | $71 \pm 4$ | $88 \pm 3$ | $\mathbf{95} \pm 1$ | $\mathbf{96} \pm 1$ | $\mathbf{96} \pm 1$ | $\mathbf{97} \pm 1$ | $\mathbf{97} \pm 2$ |
| | antmaze-large-navigate-v0 | $24 \pm 2$ | $16 \pm 5$ | $34 \pm 4$ | $75 \pm 6$ | $83 \pm 4$ | $\mathbf{91} \pm 2$ | $\mathbf{92} \pm 1$ | $90 \pm 3$ | $90 \pm 3$ |
| | antmaze-giant-navigate-v0 | $0 \pm 0$ | $0 \pm 0$ | $0 \pm 0$ | $14 \pm 3$ | $16 \pm 3$ | $65 \pm 5$ | $\mathbf{77} \pm 4$ | $73 \pm 4$ | $\mathbf{78} \pm 8$ |
| **cube** | cube-single-noisy-v0 | $8 \pm 3$ | $71 \pm 9$ | $\mathbf{99} \pm 1$ | $25 \pm 6$ | $38 \pm 2$ | $41 \pm 6$ | $33 \pm 4$ | $77 \pm 4$ | $\mathbf{97} \pm 3$ |
| | cube-double-noisy-v0 | $1 \pm 1$ | $14 \pm 3$ | $23 \pm 3$ | $3 \pm 1$ | $2 \pm 1$ | $2 \pm 1$ | $4 \pm 2$ | $19 \pm 2$ | $\mathbf{54} \pm 5$ |
| | cube-triple-noisy-v0 | $1 \pm 1$ | $9 \pm 1$ | $2 \pm 1$ | $1 \pm 0$ | $3 \pm 1$ | $2 \pm 1$ | $2 \pm 1$ | $17 \pm 3$ | $\mathbf{42} \pm 3$ |
| **scene** | scene-play-v0 | $5 \pm 1$ | $42 \pm 4$ | $51 \pm 4$ | $5 \pm 1$ | $19 \pm 2$ | $38 \pm 3$ | $20 \pm 4$ | $63 \pm 6$ | $\mathbf{78} \pm 7$ |

losses. We further compare against two recent offline GCRL methods, Option-aware Temporally Abstracted Value learning (OTA) (Ahn et al., 2026), which incorporates option-aware temporal abstraction into value learning to obtain better high-level policies, and Subgoal Advantage-Weighted Policy Bootstrapping (SAW) (Zhou & Kao, 2026), which trains a flat goal-conditioned policy by bootstrapping from subgoal-conditioned policies. Against these diverse baselines, we evaluate CoGHP's performance to demonstrate the effectiveness of our approach. Complete hyperparameter settings, including the number of latent subgoals generated by CoGHP, are provided in Appendix B. We include additional experiments, including hyperparameter ablation studies, in Appendix C.

## 5.2. Results Analysis

**Navigation Performance**   In the navigation benchmarks (pointmaze and antmaze in Table 1), CoGHP demonstrates superior performance across all task complexities, particularly excelling in the most challenging scenarios that require extensive multi-stage reasoning. On the giant maze variants, CoGHP achieved 79% on pointmaze-giant-navigate and 78% on antmaze-giant-navigate, significantly outperforming HIQL (46% and 65% respectively) and remaining competitive with or stronger than recent baselines such as OTA and SAW. This substantial performance gap highlights the limitations of HIQL's two-level hierarchical structure with separate networks when faced with tasks requiring coordination of multiple intermediate decisions. Although OTA improves hierarchical learning through temporally abstracted value estimation and SAW simplifies control by flattening the hierarchy, CoGHP retains explicit intermediate decomposition within a unified autoregressive policy, which can be beneficial for complex navigation tasks. Compared with single-subgoal hierarchical abstractions, CoGHP can flexibly generate a sequence of intermediate latent subgoals when the task benefits from explicit multi-stage decompo-

sition, enabling more sophisticated navigation planning for complex maze environments.

**Manipulation Performance**   CoGHP's advantages become even more pronounced in manipulation tasks (cube and scene in Table 1), where sequential decision-making directly benefits from our unified sequence generation approach. Scene tasks require learning complex behavioral sequences, in which agents must coordinate up to eight sequential atomic behaviors in the correct order. On the scene task, CoGHP achieved 78% compared to HIQL's 38%, OTA's 20%, and SAW's 63%, demonstrating how CoGHP's chain-of-goals approach enables effective decomposition of complex sequential tasks. Cube manipulation tasks present a different challenge, requiring repetitive pick-and-place operations where behavioral complexity is lower but precise motor control and correct placement ordering become critical. In these environments, HIQL exhibits performance degradation as the low-level policy lacks sufficient access to information about the final goal. This limitation becomes evident when comparing HIQL (41%) to GCIQL (99%) on cube-single, where HIQL loses awareness of the final goal and may select sub-optimal actions, undermining accurate cube placement. CoGHP addresses this fundamental issue through its unified optimization framework, achieving 97% on cube-single and maintaining strong performance even on the complex cube-triple (42%), outperforming OTA and SAW by a large margin on multi-object manipulation tasks. This demonstrates CoGHP's ability to retain final-goal awareness throughout the decision sequence while ensuring precise action execution, enabling successful manipulation across tasks of varying complexity.

## 5.3. Architectural Component Analysis

To validate our architectural choices, we conducted ablation studies comparing CoGHP with two variants. We consider (i) a Transformer-backbone version that replaces the MLP-Mixer blocks while keeping all other components fixed, and

*Table 2.* **Ablation Results on Architecture Variants.** Bold values indicate the highest performance values within a 5-point range of the maximum in each row. Standard deviations across 8 random seeds are shown. The experimental evidence shows that MLP-Mixer outperforms Transformer as an architectural foundation. Furthermore, the results highlight the important role of the causal mixer in achieving this performance.

| Environment | Transformer | CoGHP w/o causal mixer | CoGHP (Ours) |
|---|---|---|---|
| antmaze-medium-navigate-v0 | $97 \pm 1$ | $97 \pm 1$ | $97 \pm 2$ |
| antmaze-giant-navigate-v0 | $66 \pm 4$ | $71 \pm 7$ | $78 \pm 8$ |
| cube-single-noisy-v0 | $19 \pm 2$ | $95 \pm 4$ | $97 \pm 3$ |
| cube-double-noisy-v0 | $11 \pm 2$ | $44 \pm 4$ | $54 \pm 5$ |
| cube-triple-noisy-v0 | $2 \pm 1$ | $27 \pm 6$ | $42 \pm 3$ |

*Figure 4.* **Subgoal Count Analysis.** We compare the impact of the number of subgoals in our framework (with 8 different random seeds). Dark lines represent the average returns, and shaded areas represent standard deviations.

(ii) an MLP-Mixer version with the causal mixer removed. The results in Table 2 show that performance differences grow with task complexity. In simpler environments like antmaze-medium-navigate (97% for all variants), the choice of backbone architecture shows minimal impact, suggesting that basic sequential reasoning may be sufficient. However, as task complexity increases, MLP-Mixer provides clear advantages over Transformers, with performance gaps widening substantially in challenging scenarios like cube-triple (42% vs 2%) and antmaze-giant-navigate (78% vs 66%). This aligns with our observation that, in offline hierarchical RL where tokens have fixed, position-dependent semantic roles, Transformer backbones offer limited generalization benefits and often exhibit reduced training stability. Similarly, the causal mixer component shows minimal contribution in simple tasks (antmaze-medium and cube-single), but becomes increasingly critical as the demands for hierarchical reasoning grow. In complex manipulation tasks requiring precise sequential coordination, the causal mixer yields substantial gains on cube-triple, improving performance from 27% to 42%. This highlights its critical role in autoregressive generation by allowing each subgoal prediction to incorporate information from previously generated tokens. Further analysis of the Transformer baseline is provided in Appendix A.4, and implementation details are provided in Appendix B.3.

### 5.4. Subgoal Count Analysis

We further analyze how the subgoal count $H$ affects performance across task types. As shown in Figure 4, the optimal choice of $H$ depends on the task. In navigation environments, intermediate subgoals play an important role, as $H = 0$ performs poorly in both antmaze-large and antmaze-giant, while moderate subgoal counts achieve strong performance. However, increasing the number of subgoals is not always beneficial, as antmaze-large degrades with $H = 10$ and antmaze-giant performs best with $H = 2$. In contrast, manipulation tasks perform better with much smaller subgoal counts. Both cube-double and scene achieve the best performance with $H = 1$, with larger subgoal counts consistently performing worse. This indicates that excessive hierarchical decomposition can interfere with the precise control required for manipulation tasks. Overall, these results show that the appropriate subgoal horizon is task-dependent rather than simply improving with more subgoals.

### 5.5. Subgoal Visualizations

We visualize latent subgoals in the antmaze-giant environment to examine how CoGHP's generated subgoal chain guides the agent. To map latent subgoals from the model's latent space back to the observation space, we introduce a subgoal decoder and train it jointly with the hierarchical policy. Specifically, we add an L2 reconstruction loss between each decoded subgoal and its corresponding tar-

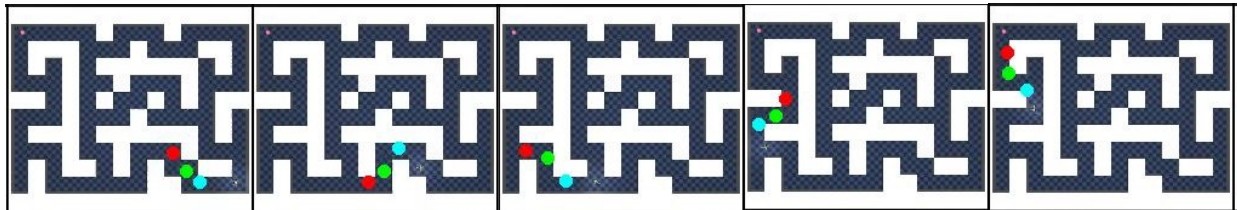

*Figure 5.* **Subgoal Visualization.** An agent located in the bottom-right corner is tasked with reaching the goal in the top-left. Here, the policy outputs three latent subgoals, plotted as blue, green, and red dots, ordered from nearest to farthest relative to the agent. The full version is in Appendix C.9.

get subgoal from the dataset. For visualization, we extract only the $x$ and $y$ coordinates of the decoded subgoals. We configure the model to generate three subgoals, and Figure 5 shows the resulting trajectories. All three subgoals lie on or near the optimal path, supporting our hypothesis that CoGHP can effectively generate multiple intermediate goals to reach the final objective. Moreover, because subgoals are generated autoregressively, the prediction of the subgoal closest to the current state is conditioned on previously generated subgoals, helping the agent produce effective actions. To further visualize subgoal generation under high-dimensional (pixel-based) observations, we additionally perform subgoal visualization on visual-antmaze. Full results for both antmaze-giant and visual-antmaze are included in Appendix C.9.

## 6. Conclusion

We introduced the Chain-of-Goals Hierarchical Policy (CoGHP), which brings the chain-of-thought-style reasoning into offline goal-conditioned RL. CoGHP tackles key limitations of earlier hierarchical methods, such as relying on a single intermediate subgoal, losing awareness of the final goal when subgoals are erroneous, and lacking end-to-end optimization. To address these limitations, CoGHP reformulates hierarchical decision-making as autoregressive sequence modeling within a unified framework. CoGHP autoregressively generates a sequence of latent subgoals followed by the primitive action within a unified model, where each latent subgoal acts as a reasoning step that conditions subsequent predictions. We introduce the use of the MLP-Mixer architecture in hierarchical RL, enabling efficient cross-token communication and learning structural relationships that support hierarchical reasoning. Experiments on challenging benchmarks show that CoGHP consistently outperforms strong baselines, demonstrating its effectiveness for long-horizon offline control. Future work may explore adaptive mechanisms that adjust the number of subgoals based on task complexity and investigate more abstract forms of subgoal representation beyond encoded future states to further improve expressiveness and generalization.

## Impact Statement

In this study, we address fundamental challenges in long-horizon offline RL by reformulating hierarchical decision-making as a structured reasoning process. By enabling reliable goal-conditioned behavior in complex environments, this work contributes to the broader development of autonomous systems and decision-making frameworks. While the advancement of such algorithms has various societal implications, particularly in the fields of robotics and automation, there are no specific ethical concerns or negative consequences that we feel must be highlighted here.

## Acknowledgements

This work was supported by the Technology Innovation Program (or Industrial Strategic Technology Development Program-Technology Innovation Program) (RS-2025-25449157, Development of Commercialization Technologies for End-to-End Autonomous Driving Products) funded by the Ministry of Trade, Industry and Resources (MOTIR, Korea), the National Research Foundation of Korea (NRF) grant funded by the Korea government (MSIT) (RS-2026-25497111), the Institute of Information & Communications Technology Planning & Evaluation (IITP) under the Artificial Intelligence Convergence Innovation Human Resources Development (IITP-2026-RS-2023-00255968) grant funded by the Korea government (MSIT), and the Ajou University research fund.

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

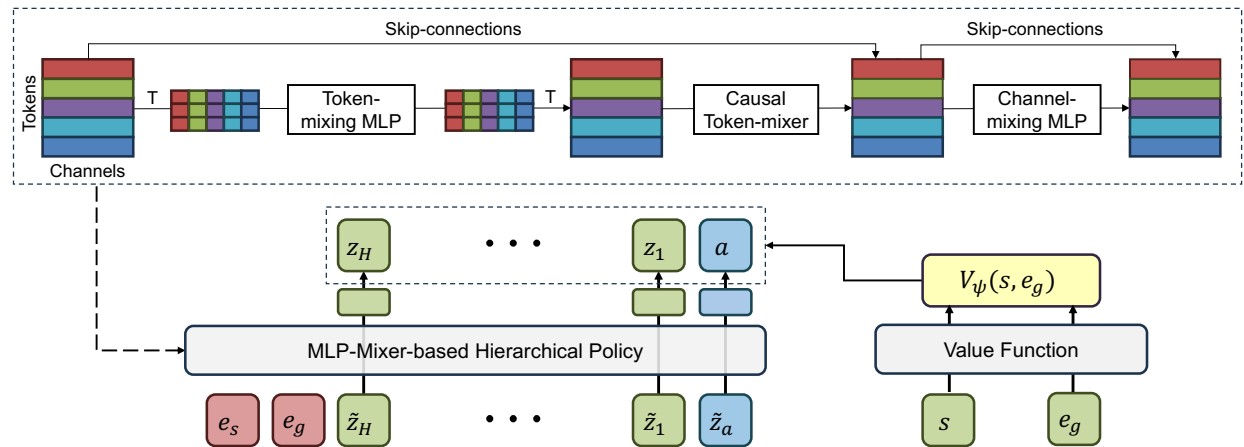

*Figure 6.* **CoGHP Architecture.** The framework comprises (1) an MLP-Mixer-based hierarchical policy that implements sequence generation for hierarchical control, autoregressively generating latent subgoals $z_H, \ldots, z_1$ ordered from farthest to nearest, and the primitive action $a$, and (2) a shared goal-conditioned value function $V_\psi(s, e_g)$ providing unified training signals for both subgoal generation and action prediction. The MLP-Mixer-based hierarchical policy takes H+3 tokens as input and processes them through alternating token-mixing, causal token-mixing, and channel-mixing layers to generate the output sequence.

## A. Algorithmic Details

### A.1. Architecture Details

#### A.1.1. OVERVIEW

MLP-Mixer enables aggregation of both global and local features across tokens without complex attention. This ability to combine information from every token makes it an ideal backbone for our hierarchical policy, which must reason over an entire sequence of latent subgoals. Building on this insight, at each time step $t$, our hierarchical policy autoregressively generates a sequence of $H$ latent subgoals and one primitive action. During subgoal prediction, CoGHP sequentially predicts latent subgoals from $z_H$, the furthest from the current state, toward $z_1$, the closest to the current state. Internally, our hierarchical policy maintains a fixed-length sequence of $T = 2 + H + 1$ tokens (two for the state and goal embeddings, $H$ for subgoal placeholders, and one for the action placeholder) and processes them through a single "Token-Mixer, Causal token-mixer, Channel-Mixer" block (Figure 6). This block shares parameters across all $H + 1$ prediction steps, enabling end-to-end gradient flow and parameter efficiency.

$$\text{input}_i = \begin{cases} (e_s, \, e_g, \, \tilde{z}_{1:H}, \, \tilde{z}_a) & i = H \\ (e_s, \, e_g, \, z_{i+1:H}, \, \tilde{z}_{1:i}, \, \tilde{z}_a) & i = H-1, \cdots, 1 \\ (e_s, \, e_g, \, z_{1:H}, \, \tilde{z}_a) & i = 0 \end{cases} \tag{10}$$

#### A.1.2. FORWARD PASS THROUGH THE MODIFIED MIXER

At each prediction step, the token sequence is first transposed and passed through the token-mixing MLP. It is then transposed back and multiplied by our learnable lower-triangular causal mixer, which enforces each token to integrate information from itself and all preceding tokens. The resulting token sequence is combined with the original input tokens via a skip connection. Next, it passes through the channel-mixing MLP, followed by another skip connection with the previous outputs. While both MLP blocks remain identical to those in the original Mixer, we introduce the causal mixer to impose sequential order and learnable inter-token dependencies.

Here, we illustrate with an example of four tokens $(e_o, e_g, z_1, z_a)$, showing how a learnable lower-triangular causal token-mixer is applied over them. First, let the token-mixing MLP produce per-token vectors

$$Y = \begin{pmatrix} y_1 \\ y_2 \\ y_3 \\ y_4 \end{pmatrix} \tag{11}$$

where $y_1 \leftrightarrow e_o$, $y_2 \leftrightarrow e_g$, $y_3 \leftrightarrow z_1$, $y_4 \leftrightarrow z_a$. We define a learnable $4 \times 4$ matrix

$$M = \begin{pmatrix} a_{11} & 0 & 0 & 0 \\ a_{21} & a_{22} & 0 & 0 \\ a_{31} & a_{32} & a_{33} & 0 \\ a_{41} & a_{42} & a_{43} & a_{44} \end{pmatrix}, \tag{12}$$

where each $a_{mn}$ (for $m \geq n$) is a trainable scalar and all entries above the diagonal are zero to block "future" tokens. We then compute the outputs as $Y' = M Y$, which component-wise yields

$$\begin{aligned} y_1' &= a_{11}\, y_1, \\ y_2' &= a_{21}\, y_1 + a_{22}\, y_2, \\ y_3' &= a_{31}\, y_1 + a_{32}\, y_2 + a_{33}\, y_3, \\ y_4' &= a_{41}\, y_1 + a_{42}\, y_2 + a_{43}\, y_3 + a_{44}\, y_4. \end{aligned} \tag{13}$$

Put simply, the lower-triangular causal mixer ensures that each token's representation is computed from its own features and those of all preceding tokens.

## A.2. Training Details

### A.2.1. GOAL DISTRIBUTIONS

We use a mixture of three goal sources when training our value function. At each update, the goal $g$ is drawn with probability 0.2 from the current state $s_t$, with probability 0.5 from a future state sampled according to $\mathrm{Geom}(1-\gamma)$, and with probability 0.3 from a uniformly chosen random state in the dataset. This combination balances learning from immediate rewards, long-horizon returns, and broad coverage of the state space. This sampling strategy follows approaches from (Ghosh et al., 2023) and (Park et al., 2023). We train the value function using these sampled states and goals via:

$$\mathcal{L}_V(\psi) = \mathbb{E}_{(s,s',g)\sim\mathcal{D}} \left[ L_2^\tau \left( r(s,g) + \gamma V_{\bar{\psi}}(s', \phi_{\psi_g}(g)) - V_\psi(s, \phi_{\psi_g}(g)) \right) \right]. \tag{14}$$

To generate training targets for CoGHP, we first sample a trajectory of length $T$ and pick a time index $t$. We uniformly sample the final goal $g$ from that trajectory. Next, we sample $H$ subgoals at fixed $k$-step intervals along the sampled trajectory. The $i$-th subgoal is the state $s_{\min(t+ik,\,T)}$, so that the policy sees subgoals starting from the farthest point $s_{\min(t+Hk,\,T)}$ and stepping inward by $k$ each prediction step until $s_{\min(t+k,\,T)}$.

## A.3. Algorithm

Algorithm 1 provides pseudocode for CoGHP.

## A.4. Transformer-baseline Analysis

As shown in Section 5.3, CoGHP achieves higher performance than the transformer-based baseline in most environments. We interpret the performance difference between the Transformer baseline and CoGHP as largely stemming from how each architecture handles position-dependent tokens. In CoGHP, unlike text in LLMs where tokens have context-dependent meanings and roles, the input sequence is composed of structured position-dependent token roles, where each index has a fixed semantic function as 'current state, final goal, sequential intermediate subgoals, and primitive action.' In such settings, prior time-series studies (Zeng et al., 2023; Chen et al., 2023) have observed that when the underlying signal is governed mainly by fixed position-dependent structure rather than rich context-dependent interactions across covariates,

---

**Algorithm 1** Chain-of-Goals Hierarchical Policy (CoGHP)

---

**Require:** offline dataset $\mathcal{D}$
 1: Initialize value function $V_\psi$, hierarchical policy $\pi_\theta$, state encoder $\phi_{\theta_o}$, goal encoder $\phi_{\psi_g}$, subgoal head $\phi_{\theta_s}$ and action head $\phi_{\theta_a}$     ($\{\psi_g\} \in \psi$, $\{\theta_o, \theta_s, \theta_a\} \in \theta$)
 2: Initialize learning rates $\eta_\psi, \eta_\theta$
 3: **for** each training iteration $n$ **do**
 4:     Sample $(s, s', g) \sim \mathcal{D}$
 5:     Compute loss $\mathcal{L}_V(\psi)$ using Equation 14
 6:     $\psi \leftarrow \psi - \eta_\psi \nabla_\psi \mathcal{L}_V(\psi)$
 7:     Sample $(s, a, s', s_{1:H}, g) \sim \mathcal{D}$
 8:     **for** each subgoal prediction step $i = H, \dots, 1$ **do**
 9:         Predict latent subgoal $z_i = \pi_\theta\big(z_i \mid e_o, e_g, z_{i+1:H}, \tilde{z}_{1:i}, \tilde{z}_a\big)$
10:         Compute hierarchical objective $J^{h_i}(\theta)$ using Equation 7
11:     **end for**
12:     Predict primitive action $a = \pi_\theta\big(a \mid e_o, e_g, z_{1:H}, \tilde{z}_a\big)$
13:     Compute low-level objective $J^\ell(\theta)$ using Equation 8
14:     Compute total objective $J_{\text{total}}(\theta)$ using Equation 9
15:     $\theta \leftarrow \theta + \eta_\theta \nabla_\theta J_{\text{total}}(\theta)$
16: **end for**

---

multivariate Transformer models can suffer from overfitting and degraded generalization, whereas time-step-dependent linear or MLP-based models tend to remain more robust. These results suggest that when token roles are relatively fixed and the signal is primarily position-dependent, the additional flexibility of data-dependent self-attention does not necessarily yield better generalization, making an MLP-Mixer backbone a natural architectural choice. Since the token roles are clearly fixed in CoGHP, this structural property helps explain the empirical Mixer-Transformer performance gap (Table 2).

## B. Implementation Details

### B.1. Environment Details

We evaluated CoGHP on a subset of OGBench (Park et al., 2024) environments covering both navigation and manipulation challenges. For navigation, experiments took place in the pointmaze and antmaze domains, where the agent must traverse from a random start to a random goal within the mazes. Each domain includes medium, large, and giant variants to progressively test long-horizon reasoning. In pointmaze, a 2D point-mass agent operates in a two-dimensional state-action space, whereas antmaze uses the same maze layouts to challenge a quadrupedal ant agent with a 29-dimensional observation space and an 8-dimensional action space.

Manipulation tasks employ a 6-DoF UR5e arm with a Robotiq 2F-85 gripper in the cube and scene scenarios. In the cube environments, the agent arranges one to three cubes into a target configuration using pick-and-place, stacking, or swapping actions. Single-cube trials have a 28-dimensional observation space, double-cube trials have a 37-dimensional observation space, and triple-cube trials have a 46-dimensional observation space. All cube environments use a 5-dimensional action space corresponding to displacements in x position, y position, z position, gripper yaw, and gripper opening. The scene setup increases the observation space to 40 dimensions, which captures object poses and lock states, while retaining a 5-dimensional action space.

All environments use a sparse reward structure where the agent receives a reward of 0 upon successfully reaching the goal and -1 at each timestep when the goal has not been reached. Following OGBench's evaluation protocol, we tested five predefined state-goal pairs per environment and reported the average success rate for both navigation and manipulation tasks.

### B.2. Hyperparameters

We categorize our experimental environments into navigation, manipulation, and visual tasks, and summarize their hyperparameters in Table 3. For the navigation tasks, the values in $\{\}$ denote the hyperparameters for the medium, large, and giant map sizes, respectively.

*Table 3.* CoGHP hyperparameters.

| Hyperparameter | Navigation | Manipulation | Visual-tasks |
|---|---|---|---|
| # gradient steps | 1000000 | 1000000 | 500000 |
| Batch size | | 256 | |
| Value MLP dimensions | | (512, 512, 512) | |
| Encoder MLP dimensions | | (512, 512, 512) | |
| Pixel-based Representation | | Impala CNN | |
| Head MLP dimensions | | (512, 512, 512) | |
| State/goal embedding dimensions | {32, 32, 128} | 256 | 32 |
| Token-mixer MLP dimensions | | (32, 32) | |
| Channel-mixer MLP dimensions | | (32, 32) | |
| # Subgoals $H$ | {1, 2, 2} | 1 | 1 |
| Weight coefficients $\lambda_h$ | {0.04, 0.02, 0.02} | 0.1 | 0.04 |
| Weight coefficients $\lambda_\ell$ | | 1 | |
| Subgoal discount factor $\gamma_h$ | | 0.8 | |
| subgoal step $k$ | {25, 50, 50} | 10 | 25 |
| Advantage temperature $\beta$ | | 3.0 | |
| Learning rate | | 0.0003 | |
| Nonlinearity | | GELU | |
| Optimizer | | Adam | |

*Table 4.* **Experimental results on pixel-based environments.** Bold values indicate the highest performance values within a 5-point range of the maximum in each column. Standard deviations across 4 random seeds are shown.

| Algorithm | visual-antmaze-medium-navigate-v0 | visual-cube-single-noisy-v0 |
|---|---|---|
| GCBC | $11 \pm 2$ | $14 \pm 3$ |
| GCIVL | $22 \pm 2$ | $75 \pm 3$ |
| GCIQL | $11 \pm 1$ | $48 \pm 3$ |
| QRL | $0 \pm 0$ | $10 \pm 5$ |
| CRL | $\mathbf{94} \pm 1$ | $39 \pm 30$ |
| HIQL | $\mathbf{93} \pm 1$ | $\mathbf{99} \pm 0$ |
| CoGHP (ours) | $\mathbf{95} \pm 2$ | $\mathbf{98} \pm 1$ |

## B.3. Transformer Baseline

The Transformer baseline uses two layers, and the token dimension is matched to CoGHP's state-embedding dimension in all environments. The parameter counts are also closely aligned; for example, on antmaze-giant, the Transformer has 5.61M parameters and CoGHP has 5.54M parameters. Both models are trained with the same optimizer, learning rate schedule, and number of training steps. Under these settings, the Mixer-Transformer comparison in this work is fair and capacity-matched with respect to both model size and training configuration.

## C. Additional Experiments

### C.1. Pixel-based Environments

We evaluated CoGHP on two additional OGBench benchmarks to test its versatility across pixel-based tasks. First, in visual-antmaze-medium, the agent receives only $64 \times 64 \times 3$ RGB frames from a third-person perspective and must infer its position and orientation by parsing the maze floor's colored tiles rather than relying on raw coordinate inputs. This pixel-only task probes CoGHP's ability to learn robust visual representations and control under perceptual uncertainty. Second, the visual-cube environment follows the same visual setup as visual-antmaze, where the agent receives only $64 \times 64 \times 3$ RGB frames from a third-person perspective. However, the manipulation arm is made transparent to ensure full observability of the object configurations and workspace.

*Table 5.* **Sensitivity Analysis of Joint Variation of $H$ and $k$.** Success rate (%) with standard deviation across 8 seeds.

*(a)* antmaze-large-navigate-v0

|         | $k = 10$     | $k = 50$    | $k = 100$   |
|---------|--------------|-------------|-------------|
| $H = 1$ | $59 \pm 11$  | $90 \pm 1$  | $86 \pm 3$  |
| $H = 2$ | $61 \pm 4$   | $90 \pm 3$  | $92 \pm 1$  |
| $H = 5$ | $41 \pm 4$   | $92 \pm 1$  | $81 \pm 4$  |

*(b)* antmaze-giant-navigate-v0

|         | $k = 10$     | $k = 50$    | $k = 100$    |
|---------|--------------|-------------|--------------|
| $H = 1$ | $10 \pm 2$   | $66 \pm 7$  | $44 \pm 10$  |
| $H = 2$ | $32 \pm 7$   | $78 \pm 8$  | $44 \pm 5$   |
| $H = 5$ | $33 \pm 10$  | $61 \pm 7$  | $53 \pm 6$   |

*(c)* cube-double-noisy-v0

|         | $k = 5$     | $k = 10$    | $k = 20$   |
|---------|-------------|-------------|------------|
| $H = 1$ | $53 \pm 11$ | $54 \pm 5$  | $32 \pm 4$ |
| $H = 2$ | $16 \pm 4$  | $27 \pm 5$  | $8 \pm 2$  |
| $H = 5$ | $8 \pm 6$   | $5 \pm 4$   | $7 \pm 3$  |

*(d)* scene-play-v0

|         | $k = 5$     | $k = 10$     | $k = 20$    |
|---------|-------------|--------------|-------------|
| $H = 1$ | $78 \pm 5$  | $78 \pm 7$   | $74 \pm 7$  |
| $H = 2$ | $1 \pm 1$   | $8 \pm 7$    | $50 \pm 9$  |
| $H = 5$ | $30 \pm 8$  | $29 \pm 15$  | $23 \pm 11$ |

*Table 6.* **Ablation Results on Subgoal Generation Order and Horizon.** Performance is reported as success rate (%) with standard deviation across 8 random seeds.

| Environment                | forward (H=2) | forward (H=5) | reverse (H=2) | reverse (H=5) |
|----------------------------|---------------|---------------|---------------|---------------|
| antmaze-large-navigate-v0  | $90 \pm 2$    | $92 \pm 2$    | $90 \pm 3$    | $92 \pm 2$    |
| antmaze-giant-navigate-v0  | $71 \pm 2$    | $50 \pm 5$    | $78 \pm 8$    | $61 \pm 7$    |

Table 4 reports CoGHP's performance alongside six benchmark methods on the visual-antmaze-medium and visual-cube-single tasks. On visual-antmaze-medium, CoGHP achieves 95% average success while CRL and HIQL attain 94% and 93% respectively, demonstrating that CoGHP retains robust goal-conditioned control under pure pixel observations. On visual-cube-single, CoGHP achieves a 98% success rate, comparable to HIQL's 99% performance and significantly outperforming other methods. These results demonstrate that CoGHP can effectively extend to tasks requiring pixel-based observations, maintaining its advantages in both navigation and manipulation tasks under visual input constraints.

## C.2. Sensitivity Analysis of Joint Variation of Subgoal Count and Subgoal Step

To analyze how the subgoal configuration influences performance, we conducted a sensitivity study where the number of generated subgoals $H$ and the subgoal step $k$ were varied jointly. We evaluated $H \in \{1, 2, 5\}$ in all settings. For navigation tasks we set $k \in \{10, 50, 100\}$, whereas for manipulation tasks we set $k \in \{5, 10, 20\}$. Table 5 summarizes the results. The navigation tasks are more sensitive to the spacing $k$ between subgoals than to the number of subgoals $H$, whereas the manipulation tasks are more sensitive to $H$. We interpret these differences as arising from task-specific characteristics. In navigation tasks, subgoals mainly serve as coarse waypoints that indicate intermediate directions or positions, so as long as they are spaced reasonably, performance is not highly sensitive to the exact number of subgoals. By contrast, in manipulation tasks, which require more precise control, an overly fine-grained subgoal chain can overconstrain the low-level policy and hinder accuracy. When $H = 1$ and the subgoal spacing is kept around $\{5, 10, 20\}$, however, performance is relatively less sensitive to $k$. This indicates that subgoal design interacts with task characteristics and supports the point made in Section D that developing methods that are robust to the choice of subgoal horizon, or can automatically select an appropriate subgoal horizon for each task, is an important direction for future work.

## C.3. Subgoal Generation Order

Our initial design assumed that subgoals closer to the current state should aggregate more comprehensive information from the hierarchical reasoning process, and we therefore generated subgoals from the one farthest from the current state to the one closest to it. To test this assumption, we add an ablation that compares forward-order generation, which generates from the subgoal closest to the current state to the farthest subgoal, against reverse-order generation. We conduct experiments on navigation tasks where generating multiple subgoals yields stable performance, and examine how environment difficulty and the number of generated subgoals $H$ affect the impact of the subgoal generation order. The results are presented in Table 6.

*Table 7.* **Ablation Results on Causal Mixer Variants.** Performance is reported as success rate (%) with standard deviation across 8 random seeds.

| Environment | w/o causal mixer | fixed causal mixer | CoGHP (Ours) |
|---|---|---|---|
| antmaze-medium-navigate-v0 | **97** ±1 | **97** ±1 | **97** ±2 |
| antmaze-giant-navigate-v0 | 71 ±7 | 72 ±1 | **78** ±8 |
| cube-single-noisy-v0 | 95 ±4 | **98** ±2 | 97 ±3 |
| cube-double-noisy-v0 | 44 ±4 | 51 ±8 | **54** ±5 |
| cube-triple-noisy-v0 | 27 ±6 | 27 ±4 | **42** ±3 |

*Table 8.* **Loss-Weight Coefficient Sensitivity.** Performance is reported as success rate (%) with standard deviation across 4 random seeds.

| Environment / Dataset | 10 | 1 | 0.1 | 0.02 | 0.01 |
|---|---|---|---|---|---|
| antmaze-giant-navigate-v0 | 0 ±0 | 2 ±1 | 66 ±4 | 79 ±8 | 65 ±4 |
| cube-double-noisy-v0 | 52 ±4 | 51 ±1 | 54 ±5 | 55 ±9 | 56 ±7 |

In the easier antmaze-large environment, forward and reverse generation perform similarly. In contrast, in antmaze-giant, reverse generation consistently outperforms forward generation, and the performance gap widens as $H$ increases. These findings support the validity of our initial design choice of generating subgoals in reverse order.

## C.4. Ablation on Causal Mixer Variants

To isolate the effect of the learnable causal mixer, we ran an ablation that compares (i) a variant that completely removes the causal mixer, (ii) a non-learnable causal mixer that replaces the learnable weights with fixed lower-triangular averaging, and (iii) default CoGHP with a learnable causal mixer (Table 7). In most environments, (i) removing the causal mixer and (ii) fixed lower-triangular averaging yield similar performance to each other, and both consistently underperform (iii) with the learnable causal mixer. This gap becomes more pronounced as task complexity increases. Thus, this experiment shows that simple causal masking or fixed averaging is not sufficient. It also indicates that a learnable causal mixer that learns the weights over past reasoning tokens plays a meaningful role in improving performance, especially on complex long-horizon tasks.

## C.5. Loss-Weight Coefficient Sensitivity

To analyze the sensitivity of our method to hyperparameter choices, we conducted comparison experiments examining the impact of the loss-weight coefficient $\lambda_h$ across different task complexities. Our analysis reveals that $\lambda_h$, which scales the subgoal generation term in Equation 9, influences training stability and performance. With a single predicted subgoal ($H = 1$, e.g., cube-double), performance is stable over a wide span of $\lambda_h$, but when two subgoals are generated ($H = 2$, e.g., antmaze-giant), setting $\lambda_h$ too high rapidly destabilizes training and drives success toward zero. Accordingly, we keep $\lambda_\ell = 1$, hold $\gamma_h$ fixed, and tune $\lambda_h$ around the heuristic value $1/k$, which balances credit assignment across the latent chain while avoiding the sharp degradation observed at larger values.

## C.6. Teacher Forcing and Subgoal Noise Ablation

To assess the role of teacher forcing and robustness to subgoal errors, we conduct two ablation studies. First, we remove teacher forcing during training, forcing the policy to rely on its own predicted subgoals. As shown in Table 9, this leads to a substantial performance drop, especially on antmaze-giant, indicating that teacher forcing is important for stable training and long-horizon rollout.

We further test robustness to subgoal prediction errors by injecting Gaussian noise into the predicted subgoal during evaluation and applying the same perturbation to HIQL. Inference-time noise causes only minor degradation in CoGHP, from 78% to 73% on antmaze-giant and from 54% to 53% on cube-double, whereas HIQL drops much more sharply on antmaze-giant. These results suggest that CoGHP is more robust to subgoal errors, likely because its unified architecture conditions action prediction on the full sequence context rather than relying on a separately generated subgoal alone.

*Table 9.* **Teacher-Forcing Ablation.** Performance is reported as success rate (%) with standard deviation across 8 random seeds.

| Environment / Dataset | HIQL-noise | HIQL | CoGHP w/o Teacher Forcing | CoGHP-noise | Ours |
|---|---|---|---|---|---|
| antmaze-giant-navigate-v0 | $33 \pm 6$ | $65 \pm 5$ | $18 \pm 5$ | $73 \pm 2$ | $78 \pm 8$ |
| cube-double-noisy-v0 | $1 \pm 1$ | $2 \pm 1$ | $3 \pm 2$ | $53 \pm 5$ | $54 \pm 5$ |

*Table 10.* **Separate Actor Ablation.** Performance is reported as success rate (%) with standard deviation across 8 random seeds.

| Environment / Dataset | CoGHP w/ Separate Actor | CoGHP (ours) |
|---|---|---|
| antmaze-giant-navigate-v0 | $56 \pm 6$ | $78 \pm 8$ |
| cube-double-noisy-v0 | $3 \pm 2$ | $54 \pm 5$ |

### C.7. Separate Actor Ablation

To examine whether CoGHP's gains arise primarily from latent subgoal generation or from its unified architecture, we compare CoGHP with a variant that uses the same subgoal-generation model but employs a separate low-level actor for primitive action prediction. As shown in Table 10, this decoupled design substantially degrades performance, reducing success from 78% to 56% on antmaze-giant and from 54% to 3% on cube-double. These results suggest that CoGHP's gains depend not only on latent subgoals, but also on the unified end-to-end design that jointly optimizes subgoal generation and action prediction.

### C.8. Robustness to Data Quality

To further examine the robustness of CoGHP to data quality, we consider two lower-quality settings: antmaze-explore datasets with weak goal-directed trajectory structure, and scene-noisy datasets with substantial action-level perturbations. As shown in Tables 11, CoGHP consistently outperforms the baselines under both settings. However, its performance still drops on the explore datasets, suggesting that weak trajectory-level structure limits the intermediate information needed to learn useful subgoals. In contrast, CoGHP is relatively more robust to action-level noise, maintaining 60% success on scene-noisy while continuing to outperform both GCIQL and HIQL. These results show that CoGHP remains competitive under lower-quality data conditions, but its performance still depends on the quality of the trajectory structure available in the offline dataset.

### C.9. Subgoal Visualizations

This subsection presents an extended version of the subgoal visualization analysis from the main text. This extended sequence offers insight into how CoGHP's autoregressive subgoal generation guides the agent through complex navigation tasks. In the antmaze-giant environment, we decode multiple latent subgoal sequences into coordinates to examine how they are positioned and what roles they play, and in the pixel-based visual-antmaze environment, we decode latent subgoals into images to verify that CoGHP produces meaningful subgoals even under more complex observations.

In the antmaze-giant environment, the results (Figure 7) show that while the farthest subgoal (red dot) sometimes positions itself in unreachable locations such as walls, the nearest subgoal (blue dot) consistently provides the agent with a reliably accessible intermediate destination by considering previously generated subgoals as well as the final objective. Another notable observation is how subgoals are positioned when approaching the final destination. When the final goal is sufficiently distant from the agent, the generated subgoals maintain regular spacing intervals. However, as the agent approaches the final goal, we observe a phenomenon where the subgoals begin to overlap. This demonstrates that CoGHP does not rigidly adhere to maintaining fixed intervals between subgoals, but rather generates optimal subgoals specifically tailored for the agent to successfully reach the final goal.

For the visual-antmaze environments, we decode latent subgoal embeddings into images for qualitative visualization. The decoder takes a latent vector as input, projects it with a fully connected layer into a small spatial feature map (e.g., $8 \times 8$ with multiple channels), and then applies a stack of transposed convolutions with stride 2 and ReLU activations to progressively upsample the features. A final transposed convolution produces an image with the target number of channels, and a bilinear resize step is applied to match the exact target resolution. The decoder is trained with a simple reconstruction objective

*Table 11.* **Results on Lower-Quality Datasets.** Performance is reported as success rate (%) with standard deviation across 8 random seeds.

| Environment | GCBC | GCIVL | GCIQL | QRL | CRL | HIQL | CoGHP |
|---|---|---|---|---|---|---|---|
| antmaze-medium-explore-v0 | $2 \pm 1$ | $19 \pm 3$ | $13 \pm 2$ | $1 \pm 1$ | $3 \pm 2$ | $37 \pm 10$ | $58 \pm 3$ |
| antmaze-large-explore-v0 | $0 \pm 0$ | $10 \pm 3$ | $0 \pm 0$ | $0 \pm 0$ | $0 \pm 0$ | $4 \pm 5$ | $17 \pm 4$ |
| scene-play-v0 | $5 \pm 1$ | $42 \pm 4$ | $51 \pm 4$ | $5 \pm 1$ | $19 \pm 2$ | $38 \pm 3$ | $78 \pm 7$ |
| scene-noisy-v0 | $1 \pm 1$ | $26 \pm 5$ | $26 \pm 2$ | $9 \pm 2$ | $1 \pm 1$ | $25 \pm 4$ | $60 \pm 5$ |

that combines mean-squared error (MSE) and $\ell_1$ loss between the decoded image and the target future-state image. As illustrated in Figure 8, CoGHP generates subgoals that still guide the agent toward the goal in this pixel-based setting. In visual-antmaze, the agent must infer its location from floor colors and wall layouts rather than explicit coordinates, and the decoded subgoal images reflect these cues by highlighting intermediate states that the agent should reach on the way to the goal. This shows that CoGHP can produce meaningful subgoals even when they must be expressed in a more complex image-based form rather than simple coordinate space.

## D. Limitations

Despite its effectiveness, CoGHP has several limitations. First, the number of latent subgoals and their timestep intervals during training must be predefined as hyperparameters. This can lead to task under-decomposition or over-decomposition, making the agent's performance heavily dependent on these hyperparameter choices. Second, CoGHP may struggle when applied to environments or tasks that differ significantly from the training dataset or when confronting tasks not present in the original data. Third, the current implementation instantiates subgoals exclusively as encoded future states, even though the MLP-Mixer-based backbone of CoGHP could, in principle, support alternative subgoal representations such as learned skill primitives or abstract semantic embeddings (Pertsch et al., 2021; Brohan et al., 2023). Finally, the present work considers only unidirectional autoregressive generation of the subgoal sequence and does not explore alternative subgoal generation schemes.

To address the first limitation, future work could introduce adaptive mechanisms that dynamically adjust the number of subgoals or their temporal spacing based on task complexity. Alternatively, the system could be configured to generate a sufficient number of subgoals while enabling the agent to adaptively terminate subgoal generation and directly produce primitive actions when appropriate. For the second limitation, potential solutions include developing rapid adaptation techniques that enable agents trained in one environment to quickly generalize to new environments or tasks. Another promising direction involves training on diverse, multi-domain datasets rather than learning each task individually, thereby creating more generalizable models capable of broader application across different scenarios and environments. To mitigate the third limitation, future work could augment offline datasets with additional modalities or annotations and introduce training objectives that align alternative subgoal latents (e.g., skill or language embeddings) with the future state distribution, enabling CoGHP to operate with non-state subgoals. Concretely, using learned skill primitives as subgoals would entail first learning skill embeddings and a decoder from offline play data, then defining CoGHP's subgoal tokens as these skill latents, and finally training the shared value function over the state-goal-skill space. Likewise, using language-based semantic subgoals would require language annotations for each subgoal, a pretrained language encoder, and an additional objective to align language embeddings with the future state distribution. For the fourth limitation, a natural extension is to equip CoGHP with bidirectional or diffusion-style planners in the subgoal space that refine the entire subgoal sequence while retaining the simplicity and effectiveness of the current causal formulation.

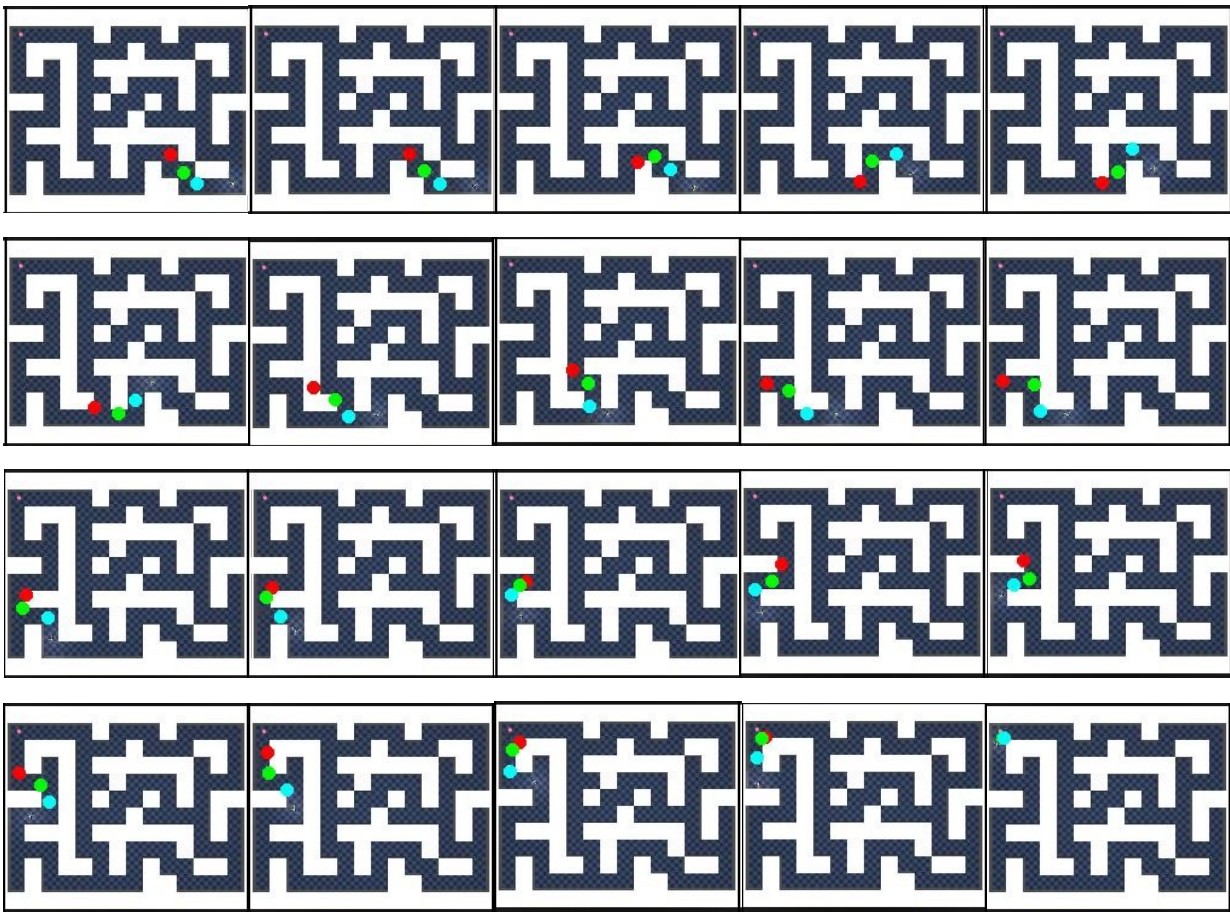

*Figure 7.* **Antmaze Subgoal Visualization.** To examine the role of CoGHP's latent subgoal chain, we decoded and visualized these subgoals in the antmaze-giant environment. In this scenario, the agent starts in the bottom-right corner and must reach the goal in the top-left. For this example, we configured the policy to output three latent subgoals, which we plotted as colored dots in blue, green, and red, ordered from nearest to farthest from the agent.

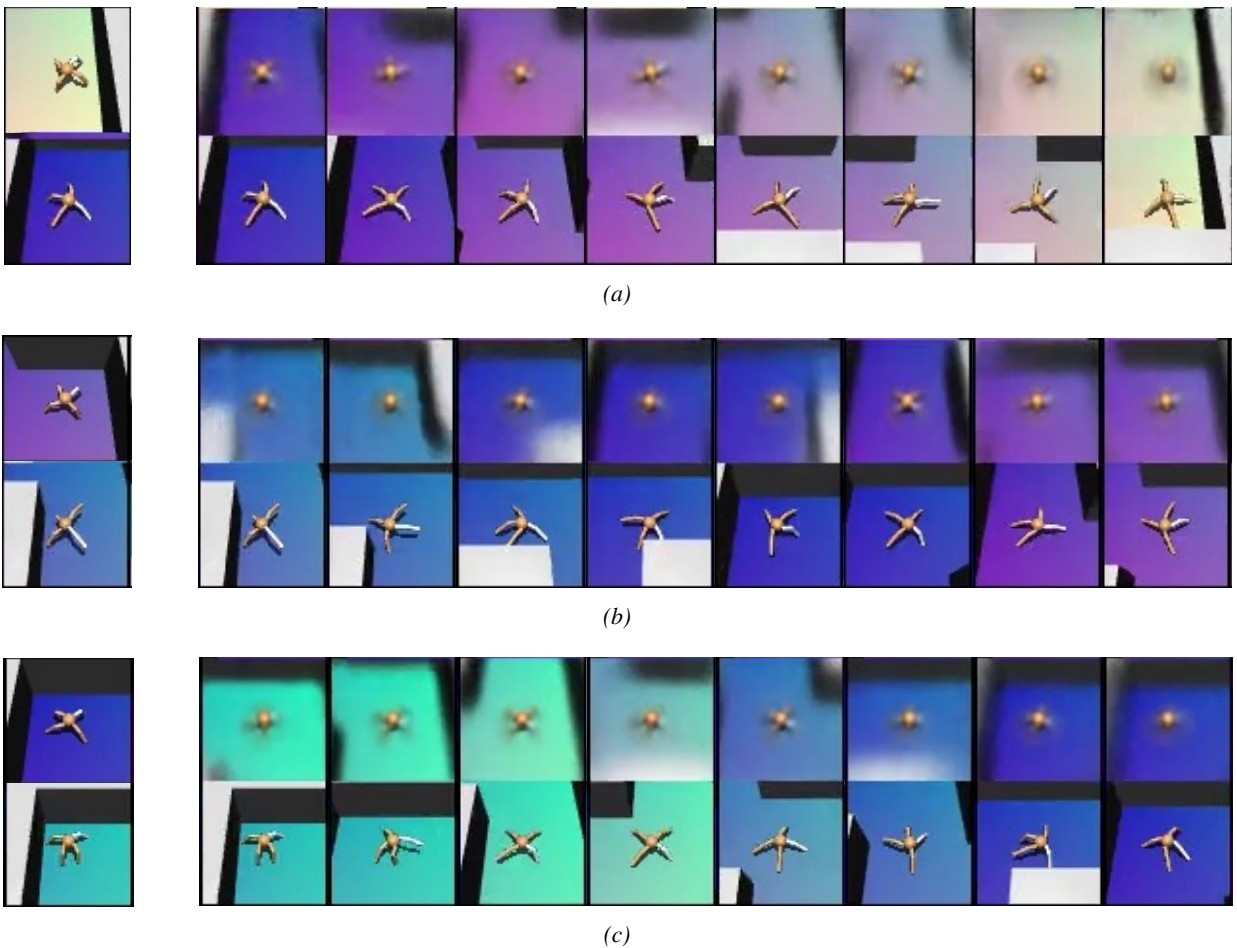

*Figure 8.* **Visual-Antmaze Subgoal Visualization.** We decoded and visualized the generated subgoals in the visual-antmaze environment. In the leftmost image, the lower panel shows the initial state, and the upper panel shows the goal state.

