# OpenReview forum: "Chain-of-Goals Hierarchical Policy for Long-Horizon Offline Goal-Conditioned RL"
_ICML.cc/2026/Conference — ICML 2026 regular_

### Official Review · Reviewer_1pQq · 2026-03-02

**Soundness:** 3
**Presentation:** 3
**Significance:** 3
**Originality:** 3
**Overall Recommendation:** 4
**Confidence:** 3

**Summary:**

The paper proposes CoGHP for long-horizon offline goal-conditioned RL. It generates multiple latent subgoals autoregressively within a single unified model. It uses an MLP-Mixer backbone and is trained with a shared value function. Experiments show strong improvements over prior hierarchical and offline RL methods, especially on complex long-horizon navigation and manipulation tasks.

**Compliance With Llm Reviewing Policy:**

Affirmed.

**Final Justification:**

I recommend a weak accept: the paper is sound, clearly presented, and makes contributions with empirical support. The rebuttal addressed some concerns but did not materially change my confidence, so my overall assessment remains largely unchanged.

**Key Questions For Authors:**

- The work is interesting. I am curious how the generated latent subgoals compare to HIQL’s latent subgoals?
- Offline RL strongly depends on dataset quality and coverage. Can the authors clarify the characteristics of the offline datasets used in OGBench and whether CoGHP is robust across different data qualities?
-

**Limitations:**

Yes. The author lists the limitations in the appendix.

**Strengths And Weaknesses:**

Strengths:

- The paper is well written and clearly structured.
- It reformulates hierarchical reinforcement learning as autoregressive subgoal generation.
- The experiments show significant gains on long-horizon navigation and manipulation tasks. And the ablation shows MLP-Mixer is important.

Weaknesses
- The theoretical proof is limited; it would be better if the author could explain more.

---

> ### Author Rebuttal · Authors · 2026-03-31
>
> Thank you for the thoughtful comments. We especially appreciate the reviewer’s question regarding the role of latent subgoals, which is one of the main distinctions between CoGHP and HIQL.
>
> > **W1. The theoretical proof is limited.**
>
> Thank you for this comment. While our work does not center on a new theoretical guarantee, CoGHP’s main contribution is a new approach to hierarchical decision-making and a unified architectural framework for offline hierarchical RL. Specifically, CoGHP reformulates hierarchical decision making as autoregressive sequence generation within a unified MLP-Mixer-based architecture. The validity of this design is supported empirically by the results in `Tables 1 and 2` and the ablation studies in `Appendix C`. To address the reviewer’s concern, the final version will clarify the intuition behind these design choices and discuss their limitations more explicitly.
>
> > **Q1. How do the generated latent subgoals compare to HIQL’s latent subgoals?**
>
> The latent subgoals in HIQL and CoGHP play structurally different roles. In HIQL, the high-level policy takes the current state and the final goal as input and predicts a single subgoal in the form of an encoded future state at a fixed timestep ahead. In contrast, the latent subgoals in CoGHP are generated autoregressively within a unified architecture and function as intermediate reasoning tokens that incorporate both the final goal and the previously generated subgoals. As a result, even when CoGHP generates only a single subgoal, it is designed to preserve information about the final goal and to be used directly for action generation. By contrast, because HIQL separates the high-level and low-level policies, information about the final goal can be weakened during low-level action selection. This difference is also reflected in the empirical results. In `Table 1`, on cube-single-noisy-v0, HIQL achieves 41% whereas CoGHP achieves 97%, which suggests that CoGHP preserves final-goal awareness more effectively throughout the decision-making process.
>
> > **Q2. Characteristics of the offline datasets and robustness to data quality**
>
> OGBench is a benchmark for offline goal-conditioned RL that provides diverse navigation and manipulation environments, along with datasets of varying quality. The main experiments in this paper use relatively high-quality datasets, but to further examine how robust CoGHP is to changes in data quality, we additionally considered two lower-quality settings. First, the explore dataset consists of random exploratory trajectories and therefore contains weak goal-directed trajectory structure. Second, the noisy dataset is collected by adding Gaussian noise to an expert policy, so the overall trajectory direction is largely preserved while the action-level perturbations are substantial.
>
> On the explore dataset, CoGHP consistently outperformed HIQL, although a clear performance drop was still observed relative to the high-quality setting. This likely arises because weak goal-directed trajectory structure also limits the intermediate structural information needed for CoGHP to learn useful subgoals. In this sense, CoGHP, like other offline RL methods, is sensitive to degradation in trajectory-level data quality. In contrast, CoGHP appeared relatively more robust to action-level noise. For example, in the scene environment, CoGHP still maintained 60% performance when moving from play to noisy and continued to outperform both HIQL and GCIQL under the noisy condition. This suggests that when the overall trajectory structure is preserved, individual action noise does not completely break CoGHP’s hierarchical decision-making process.
>
> Overall, CoGHP remains competitive relative to the baselines even under lower-quality data conditions, but it is still affected by dataset quality. The current paper discusses the difficulty of generalizing to environments or tasks that differ from the training data in `Section D`. In response to the reviewer’s feedback, the final version will also add dependence on dataset quality as an important limitation.
>
> | Environment | GCIVL | HIQL | CoGHP |
> | --- | --- | --- | --- |
> | antmaze-medium-explore-v0 | 19$\pm$3 | 37$\pm$10 | 58$\pm$3 |
> | antmaze-large-explore-v0 | 10$\pm$3 | 4$\pm$5 | 17$\pm$4 |
>
> | Environment | GCIQL | HIQL | CoGHP |
> | --- | --- | --- | --- |
> | scene-play-v0 | 51$\pm$4 | 38$\pm$3 | 78$\pm$7 |
> | scene-noisy-v0 | 26$\pm$2 | 25$\pm$4 | 60$\pm$5 |

---

> > ### Author Rebuttal · Reviewer_1pQq · 2026-04-03
> >
> > We thank the authors for their response. My concerns are fully resolved, and I will keep my positive score.

---

> > > ### Author Response · Authors · 2026-04-03
> > >
> > > We are very pleased to hear that your concerns have been fully resolved. Your questions helped us further clarify and better highlight our contributions. We also sincerely appreciate your positive feedback on our work.

---

### Official Review · Reviewer_pvFD · 2026-03-06

**Soundness:** 4
**Presentation:** 3
**Significance:** 3
**Originality:** 3
**Overall Recommendation:** 5
**Confidence:** 4

**Summary:**

In this paper, the authors introduce CoGHP, an hierarchical approach for offline RL inspired by Chain-of-thought methods, thus modeling the decision problem as a sequential autoregressive problem. In particular, the authors provide two novel insights: first, from a model design point of view, the authors choose to have a single unified model to represent all hierarchies (in contrast with the two network modules of HIQL); second, from an model architecture point of view, the authors propose to replace the transformer architecture with a MLP-Mixer module, augmented with a learnable causal token-mixer. The authors extensively evaluate their approach across standard benchmarks of offline RL, highlighting how their approach outperforms other strong baselines, especially in longer-horizon tasks.

**Compliance With Llm Reviewing Policy:**

Affirmed.

**Final Justification:**

My opinion of the paper remains the same after rebuttal: well written paper, strong experimental results, clear contributions. As such, I maintain my positive accept score.

**Key Questions For Authors:**

1 - How does your method perform against the baselines when provided with lower-quality datasets, for example, with the "explore" versions of the Pointmaze and Antmaze datasets?

2 - Can you comment on the significant performance discrepancy of the HIQL baseline in the cube-single-noisy-v0 dataset, between state and vision-based inputs (Table 4)?

**Limitations:**

Yes.

**Strengths And Weaknesses:**

- **Soundness**: The work presented in this submission is of excellent soundness: the authors clearly motivate each component of their model and its training procedure. The authors present further implementation and training details in Appendix, which are most welcomed. Additionally, the claims of the paper are strongly supported by the results, especially in more complex, long-horizon scenarios. For completeness, it would be interesting to evaluate the performance of CoGHP in lower-quality datasets (like the explore versions of the pointmaze/antmaze datasets), and not just in datasets collected by high-quality policies. The authors also present extensive ablation studies that highlight the importance of their architectural and training choices, especially in longer-horizon tasks.

- **Presentation**: Overall the paper is of very high quality, easy to read, with high-quality figures and with no major typos. The authors carefully introduce the necessary background information before introducing their method. I did, however, find some descriptions quite repetitive and spread out throughly the document (for example, the description of the MLP-Mixer is slightly repeated in Section 2, Section 3.3 and Section 4.1), so I would incentivize the authors to streamline them. The authors also do a good job of positioning their work against other strong methods in offline RL, in particular against HIQL.

- **Significance**: Long-horizon decision-making, in particular in the offline RL setting, remains an open problem. This work provides further evidence in support of hierarchical approaches to address this problem, improving significantly over HIQL (the prior state of the art). As such this work extends the performance frontier of offline RL methods. However, it is still not clear how these methods would translate to real-world tasks (e.g., robotics, autonomous vehicles) where behavior cloning methods are still widely employed.

- **Originality**: The major novelty of this paper lies in the use of a single unified model for all hierarchical levels, replacing the need to have separate (and often trained independently) networks with a single, autoregressive model. The use of an MLP-Mixer module instead of a Transformer architecture is also quite insightful, especially in longer-horizon tasks.

---

> ### Author Rebuttal · Authors · 2026-03-31
>
> Thank you for the thoughtful feedback. We especially appreciate the suggestion to evaluate CoGHP on lower-quality datasets.
>
> > **W1. Repetition in the description of the MLP-Mixer**
>
> Thank you for pointing out that the description of the MLP-Mixer is repeated across `Sections 2, 3.3, and 4.1`. Through CoGHP, we aimed to introduce MLP-Mixer to offline goal-conditioned RL and to propose a new framework inspired by chain-of-thought reasoning, so we considered it important to explain the architectural choice and the operation of CoGHP. At the same time, we agree that some of this discussion is repetitive and may interrupt the flow of the paper. In the final version, we will reorganize the repeated descriptions so that the core framework narrative of CoGHP comes across more clearly.
>
> > **W2. Transferability to real-world tasks**
>
> As the reviewer noted, behavior cloning-based approaches currently show strong performance on real-world tasks. At the same time, these methods have a fundamental limitation in that they depend heavily on large amounts of high-quality task-specific demonstration data. Offline RL methods, including CoGHP, may not yet achieve the same level of real-world performance as these approaches. However, they still point to a meaningful direction by enabling the use of more general and heterogeneous offline data rather than relying only on carefully curated demonstrations for specific tasks. In particular, CoGHP provides a unified framework for hierarchical decision making and action generation in long-horizon problems, which may serve as a useful foundation for future extensions to real-world settings.
>
> > **Q1. Performance on lower-quality datasets**
>
> In addition to the explore datasets requested by the reviewer, we also compared the play and noisy datasets in the scene environment. These two settings reflect different types of data quality degradation. The explore datasets consist of random exploratory trajectories and therefore represent low-quality data at the trajectory level, where consistent goal-directed behavior is largely absent. In contrast, the noisy dataset is collected by adding uncorrelated Gaussian noise to an expert policy, so the overall trajectory direction is preserved while individual actions contain substantial noise.
>
> On the explore dataset, CoGHP consistently outperformed other baselines, although a clear performance drop was still observed relative to the high-quality setting. This likely arises because the lack of goal-directed trajectory structure limits the intermediate structural information needed for CoGHP to learn useful subgoals. In this sense, CoGHP, like other offline RL methods, is sensitive to degradation in trajectory-level data quality. In contrast, CoGHP appeared relatively more robust to action-level noise. For example, in the scene environment, CoGHP still maintained 60% performance when moving from play to noisy and continued to outperform both HIQL and GCIQL under the noisy condition. This suggests that when the overall trajectory structure is preserved, individual action noise does not completely break CoGHP’s hierarchical decision-making process.
>
> Overall, CoGHP remains competitive relative to the baselines even under lower-quality data conditions, but it is still affected by dataset quality. The current paper discusses the difficulty of generalizing to environments or tasks that differ from the training data in `Section D`. In response to the reviewer’s feedback, the final version will also add dependence on dataset quality as an important limitation.
>
> | Env | GCIVL | HIQL | CoGHP |
> | --- | --- | --- | --- |
> | antmaze-medium-explore-v0 | 19$\pm$3 | 37$\pm$10 | 58$\pm$3 |
> | antmaze-large-explore-v0 | 10$\pm$3 | 4$\pm$5 | 17$\pm$4 |
>
> | Env | GCIQL | HIQL | CoGHP |
> | --- | --- | --- | --- |
> | scene-play-v0 | 51$\pm$4 | 38$\pm$3 | 78$\pm$7 |
> | scene-noisy-v0 | 26$\pm$2 | 25$\pm$4 | 60$\pm$5 |
>
> > **Q2. Performance gap of HIQL on cube-single-noisy between state-based and vision-based inputs**
>
> In the OGBench paper, this gap is explained through the way the subgoal representation is learned. In the pixel-based setting, the policy loss provides an additional learning signal for the subgoal representation, whereas in the state-based setting this effect was not sufficient. The paper also notes that learning stable subgoal representations remains an important open problem in hierarchical RL. In comparison, CoGHP achieves consistently strong performance in both the state-based setting (97%) and the vision-based setting (98%). This suggests that CoGHP alleviates the instability of subgoal representation learning observed in HIQL by jointly optimizing subgoal generation and action prediction end-to-end within a unified architecture.

---

> > ### Author Rebuttal · Reviewer_pvFD · 2026-04-03
> >
> > The authors have addressed positively my concerns and, as such, I maintain my score.

---

> > > ### Author Response · Authors · 2026-04-04
> > >
> > > Thank you for your positive evaluation of our work. We are glad that we have addressed your concerns. Your comments helped us improve the clarity of the paper and better highlight our contributions.

---

### Official Review · Reviewer_Gj9N · 2026-03-10

**Soundness:** 2
**Presentation:** 3
**Significance:** 3
**Originality:** 3
**Overall Recommendation:** 4
**Confidence:** 3

**Summary:**

This paper proposes the Chain-of-Goals Hierarchical Policy (CoGHP) method for solving long-horizon goal-conditioned offline RL tasks. The method involves autoregressive prediction of latent subgoals (future states across fixed time intervals - waypoints) using an MLP-Mixer backbone with causal token mixing (instead of the classic Transformer backbone) for the algorithm and an IQL-style shared value function. Results on OGBench show strong performance, especially on difficult long-horizon navigation and manipulation tasks.

**Compliance With Llm Reviewing Policy:**

Affirmed.

**Final Justification:**

My main concerns have been addressed, and I increased my score from 3 to 4.

**Key Questions For Authors:**

Overall, I find the paper interesting and the proposed approach promising, with encouraging empirical results. I have a few concerns outlined above, but I am open to discussion and willing to update my score if the authors can address them in the rebuttal.

**Limitations:**

Yes

**Strengths And Weaknesses:**

**Strengths:**

1. The idea of ​​predicting waypoints in an autoregressive sequence generation paradigm is interesting

2. The results on the popular OGBench are convincing, and the average for 8 seeds was also calculated

3. The paper contains ablation with useful information for understanding the method and comparison with common transformer architecture

4. The authors have attached the code for their work


**Weaknesses:**
1. Although the paper explicitly states that the Chain-of-Thought (CoT) paradigm is used only as a motivation for the proposed method and is not explicitly used in the traditional NLP sense, this formulation is still somewhat confusing on first reading, when one expects the task decomposition into subtasks to be performed by CoT in the conventional sense of the term. Moreover, the statement (L106-107, left) "...adapts the chain-of-thought reasoning paradigm to offline hierarchical RL" is incorrect, as no CoT adaptation and no reasoning is made here; instead, a similar, inspired method is proposed. I recommend the authors more carefully rephrase all instances of CoT in the text to avoid this confusion.

2. The number of subgoals (H) and the distance between them (k) are hyperparameters that should be selected based on prior knowledge of the task. Appendix, Table 5 shows a sensitivity analysis of these two parameters, but the behavior of the method for H>>5 remains unclear (by the way, it would be better to move this analysis to the main text). It would be nice to see a plot of the SR(H) dependence for a specific task with a fixed k. Furthermore, Table 5 suggests that, in principle, a single subgoal (H=1) is sufficient for all tasks considered (even in the case of (b) antmaze-giant-navigate-v0, since the std values ​​of H=2 and H=1 intersect). This somewhat weakens the main thesis of the paper about the need for a deeper hierarchy and autoregressive generation of the subgoal chain (Figure 6 suggests that H=1 is optimal for various tasks).  In particular, it remains unclear to what extent the observed improvements are due to the ability to model multiple intermediate goals, rather than to other components of the proposed architecture.

3. During training, the model sees ground truth subgoals, but during evaluation, there's a chance that one of the subgoals will be generated inaccurately, causing subsequent subgoal predictions to increasingly diverge from the true values ​​due to the autoregressive nature of the method. It would be useful to empirically test the method's robustness to such errors, and for this purpose, I propose two experiments. Such an experiment would allow us to understand whether the model is capable of adjusting the hierarchical design in the presence of errors in intermediate subgoals, or whether errors lead to further accumulation of deviations during the autoregressive generation process:

	1) The model is trained as is currently the case in the paper; during validation, a small bias/noise is added to the first predicted subgoal. This allows us to determine whether potential inaccuracies in subgoal predictions during evaluations are significantly biasing the model.

	2) During training, slightly distorted (non-ground-truth) subgoals are periodically introduced into the subgoal sequence, for example, by replacing one of the subgoals in the chain with the model's predicted state or with a slightly biased state. This will allow us to determine whether the model is capable of learning to correct the hierarchical subgoal prediction.

4. Although Table 2 presents an analysis of the architectures, it does not highlight the contribution of the chain-of-goals itself. Specifically, all variants are still based on autoregressive subgoal prediction. It would be useful to include a baseline using the same architecture but predicting only one subgoal (H=1) without the autoregression. This would help clarify whether the observed improvements are due to the modeling of multiple intermediate subgoals or primarily to the architectural design of the policy network.

5. The statement that subgoals need to be predicted in reversed order along with Table 6 did not convince me, since in this Table forward (H=5) and reverse (H=5), as well as forward (H=2) and reverse (H=2) have intersecting standard deviations, which is why it is impossible to draw a conclusion about which option is actually better.

6. The ablations performed in this study are limited. Several features were introduced simultaneously: autoregressive hierarchy, MLP-Mixer backbone, causal mixer, shared value function, and so on. Because all features are enabled simultaneously, it is difficult to assess which provides the greatest impact.

7. What is the computational overhead of the model? How does its latency and compute cost compare to other baselines?

---

> ### Author Rebuttal · Authors · 2026-03-31
>
> Thank you for the thoughtful feedback. We especially appreciate the reviewer’s suggestion to test robustness by injecting noise into subgoals.
>
> > **W1. Confusion around the CoT terminology**
>
> As the reviewer correctly understood, CoGHP does not directly apply chain-of-thought, but is instead inspired by the idea of decomposing a complex problem into intermediate steps and solving it sequentially. We agree that phrases such as “adapts the chain-of-thought reasoning paradigm” may cause confusion. In the final version, we will revise the CoT-related wording throughout the paper with more precise expressions.
>
> > **W2. Sensitivity to the hyperparameters H**
>
> A related analysis can be found in `Appendix C.2`, where we compare H from 0 to 10 with fixed k. The results show that H=1 performs well in several environments, while the appropriate subgoal horizon is still task-dependent. At the same time, we would like to emphasize that CoGHP’s strength does not lie only in its ability to generate multiple subgoals, but also in its unified autoregressive architecture that connects latent subgoals and primitive actions while preserving final-goal awareness. This structural difference helps explain why CoGHP outperforms HIQL even in the H=1 setting.
>
> > **W3. Error accumulation during inference**
>
> Thank you for this helpful suggestion. We evaluated both directions proposed by the reviewer by using the w/o Teacher Forcing results in `Section C.8` and by adding Gaussian noise into the first predicted subgoal during evaluation, with the same test applied to HIQL.
>
> The results show that removing ground-truth subgoals causes a substantial performance drop, whereas inference-time noise causes only minor degradation in CoGHP but much larger drops in HIQL. This supports that CoGHP is more robust to subgoal errors due to its unified architecture. We will add this analysis to the final version.
>
> | Env | HIQL-noise | HIQL | CoGHP w/o Teacher Forcing | CoGHP-noise | Ours |
> | --- | --- | --- | --- | --- | --- |
> | antmaze-giant-navigate-v0 | 33$\pm$6 | 65$\pm$5 | 18$\pm$5 | 73$\pm$2 | 78$\pm$8 |
> | cube-double-noisy-v0 | 1$\pm$1 | 2$\pm$1 | 3$\pm$2 | 53$\pm$5 | 54$\pm$5 |
>
> > **W4. Unclear contribution of the chain-of-goals itself**
>
> The reviewer’s proposed baseline corresponds to a single-subgoal, non-autoregressive setting, which provides a comparison axis similar to HIQL. To further examine whether the advantage comes mainly from use of latent subgoals or from the unified framework, we additionally compared CoGHP against a variant that uses the same subgoal-generation model but a separate low-level actor. The substantial drop in both environments suggests that the gains of CoGHP are not explained solely by the use of latent subgoals, but also depend importantly on the unified end-to-end design that jointly optimizes subgoal generation and action prediction.
>
> | Env | CoGHP (separate actor) | CoGHP |
> | --- | --- | --- |
> | antmaze-giant-navigate-v0 | 56$\pm$6 | 78$\pm$8 |
> | cube-double-noisy-v0 | 3$\pm$2 | 54$\pm$5 |
>
> > **W5. Uncertainty about the benefit of reverse-order generation**
>
> We agree with the reviewer’s point. Given the overlap in the results reported in `Table 6`, it would be too strong to claim that reverse-order generation is conclusively better based on this table alone. Rather than presenting it as a necessary design choice, we will frame it more conservatively as empirical support for our design intuition.
>
> > **W6. Limited ablations**
>
> We understand the reviewer’s concern. Although the current paper does not evaluate every possible combination, it does examine the main design choices through ablations and related comparisons. In particular, the backbone, causal mixer, subgoal hierarchy, and unified end-to-end optimization are analyzed separately. We agree that these results are distributed across the main text and the appendix, which may make the role of each component less clear at a glance. In the final version, we will reorganize this part more clearly.
>
> > **W7. Missing analysis of computational cost**
>
> We measured the training time, number of parameters, inference latency, and FLOPs of HIQL and CoGHP on antmaze-giant using a single NVIDIA GeForce RTX 3090 GPU.
>
> | Method | Training Time | #Params | Inference Latency | FLOPs |
> | --- | --- | --- | --- | --- |
> | HIQL | 3 h | 3.8 M | 0.8 ms | 2.9 M |
> | CoGHP | 5 h | 5.5 M | 4.4 ms | 7.1 M |
>
> CoGHP requires more computation than HIQL, but its 4.4 ms inference latency remains practical. To test whether the gains come simply from model size, we also compared against a larger HIQL matched to CoGHP.
>
> | Env | HIQL | HIQL (Large) | CoGHP (Ours) |
> | --- | --- | --- | --- |
> | antmaze-giant-navigate-v0 | 65$\pm$5 | 67$\pm$5 | 78$\pm$8 |
> | cube-double-noisy-v0 | 2$\pm$1 | 2$\pm$1 | 54$\pm$5 |
>
> Even with a comparable parameter count, HIQL does not close the gap to CoGHP, which suggests that the gains of CoGHP are not explained by model size alone.

---

> > ### Author Rebuttal · Reviewer_Gj9N · 2026-04-02
> >
> > I thank the authors for their response and for conducting additional experiments at my request. My main concerns have been addressed, and I will increase my score from 3 to 4.

---

> > > ### Author Response · Authors · 2026-04-03
> > >
> > > We are very pleased to hear that your main concerns have been addressed, and we sincerely thank you for raising your score. We also greatly appreciate your valuable feedback and suggestions for additional experiments, which have helped us improve the paper and better highlight our contributions.

---

### Official Review · Reviewer_Dakr · 2026-03-12

**Soundness:** 2
**Presentation:** 3
**Significance:** 3
**Originality:** 3
**Overall Recommendation:** 3
**Confidence:** 5

**Summary:**

The paper claims that traditional hierarchical approaches predict only a single intermediate subgoal and argues that this is inadequate for tasks which require the coordination of multiple intermediate decisions, and propose to solve it with autoregressive sequence modeling of latent subgoals. They use GCIQL to learn a goal-conditioned value function and train an MLP Mixer architecture to predict the next (embedded) subgoal given the state, goal, and a sequence of previously generated subgoals at fixed $k$-step intervals, in addition to the immediate primitive action $a$ — importantly, this generative process takes place in reverse, starting with subgoals closest to the goal. In an extension of HIQL, which trains the high-level policy with AWR, they train the autoregressive model with AWR applied to each subgoal.

**Compliance With Llm Reviewing Policy:**

Affirmed.

**Final Justification:**

Same as stated in the response to the authors, and copied below:

I thank the authors for their detailed response. My main issue with the paper remains in the narrative - it is explicitly stated that traditional single-subgoal hierarchical approaches fail because they only predict one subgoal and that makes them inadequate for complex tasks, but this is only supported in select navigation tasks (i.e., antmaze only), with strong negative results in other environments.

While I agree that the appropriateness of subgoal decomposition is task-dependent, I maintain that the tasks that show significant degradation with more subgoals (multi-object manipulation) are exactly the sort of compositional tasks with temporal ordering that one should expect multi-subgoal planning to help with, and it is unclear why task decomposition fails here. In the end, I don't see sufficient evidence to support the role of the multi-subgoal (reverse) autoregressive generation in improving performance. Instead, based on their results in Table 5 and their results with a separate low actor, it seems (1) multiple subgoals are bad for performance and (2) that the performance benefits instead come from improved primitive actions, not temporally extended reasoning.

On a more positive note, the empirical results are strong and the aforementioned results provide some support to their claim that the unified architecture supports "final goal awareness" during the primitive action generation. Additionally, some of my original concerns about related works have been resolved, and so I will raise some components of my score. In the end, however, I find that the narrative does not align with the findings and obscures the main contributions of the paper. If rewritten, it would make a strong submission in the future.

**Key Questions For Authors:**

1. I would be curious to see if the benefits come solely through improved subgoal generation, e.g., if you were to only use CoGHP to generate subgoals, then train a low-level actor similar to HIQL conditioned on the latent subgoal embeddings.
2. Why are the experiments for visual environments limited to two simple tasks that do not test long-horizon reasoning (`antmaze-medium` and `cube-single`) and are already saturated by HIQL? The main claim of the paper seems to be the flaws of planning with only a single subgoal, so I would expect improvements over HIQL in this setting as well and improvements over one-step methods are expected.
3. As is, it is unclear if CoGHP can achieve the same sort of compositional generalization as similar generative methods such as CompDiffuser, given the lack of experiments on OGBench `stitch` or `explore` datasets - can you report some results in these settings?

**Limitations:**

Yes

**Strengths And Weaknesses:**

**Strengths**

The choice of architecture is well-justified and the empirical comparisons to Transformers are interesting. The empirical results, especially on cube-triple, are quite strong.

**Weaknesses**

The main weakness of this work is novelty and coverage of related literature. The Related Works section only includes references up to 2024, and misses many recent relevant works that directly address the problem of predicting only one subgoal [1, 2, 3, 4]. In particular, [3] also proposes to predict a sequence in reverse order starting from the goal. Following from this, the results only compare against the original baselines from OGBench and do not incorporate relevant modern baselines, given the paper’s positioning as a hierarchical/planning method.

1. **Architectural emphasis**: The choice of architecture is well-justified, although I’m not sure that it is necessary to devote so much of the paper to discussing what amounts to a simple architectural choice. While I appreciate the discussion, I think centering too much of the paper around it obscures the flow of the narrative w.r.t. the CoT contribution.
2. **Weak baselines:** The paper only uses simple one-step approaches and HIQL as baselines, where HIQL is the only hierarchical algorithm and is known to suffer from representational pathologies [5, 6]. There should be some comparisons against stronger and more recent baselines that implement forms of horizon reduction to better establish the benefits of multigoal planning, such as OTA [5] or SAW [6], the latter of which also uses a single one-step GCIVL-trained value function and a unified architecture; or even more closely related “jumpy” planning models capable of sparse parallel planning, such as CompDiffuser [1], MCTD [7] and Fast-MCTD [8].
    1. For example, the 4th limitation/future direction stated on line 1082 essentially describes CompDiffuser.
3. **Multiple subgoals seem unnecessary**: The method seems quite sensitive to the choice of $H$, and indeed using longer horizons $H=2$ seems to significantly harm performance as seen in ``cube-double-noisy`` and ``scene-play`` in Table 5. The fact that $H=1$ seems to be the universally best choice goes against the claim that predicting sequences of subgoals is necessary or even beneficial at all.

**References**
1. Luo et al. (2025). Generative Trajectory Stitching through Diffusion Composition. NeurIPS 2025.
2. Chen et al. (2025). Hierarchical Multiscale Diffuser for Extendable Long-Horizon Planning. arXiv preprint.
3. Zhang et al. (2025). Chain-of-Action: Trajectory Autoregressive Modeling for Robotic Manipulation. NeurIPS 2025.
4. Chen et al. (2024). Simple Hierarchical Planning with Diffusion. ICLR 2024.
5. Ahn et al. (2025). Option-aware Temporally Abstracted Value for Offline Goal-Conditioned Reinforcement Learning. NeurIPS 2025.
6. Zhou et al. (2025). Flattening Hierarchies with Policy Bootstrapping. NeurIPS 2025.
7. Yoon et al. (2025). Monte Carlo Tree Diffusion for System 2 Planning. ICML 2025.
8. Yoon et al. (2025). Fast Monte Carlo Tree Diffusion: 100x Speedup via Parallel Sparse Planning. NeurIPS 2025.

---

> ### Author Rebuttal · Authors · 2026-03-31
>
> Thank you for your thoughtful feedback. We appreciate the reviewer’s suggestions for additional related work.
>
> > **W0. Limited novelty and missing related literature**
>
> Thank you for recommending [1-4]. We will incorporate them into the Related Work section in the final version. Additional baselines are addressed in `W2`.
>
> We would like to emphasize that the novelty of CoGHP lies in introducing a new way to model hierarchy in offline goal-conditioned RL, by casting hierarchical decision-making as unified autoregressive latent subgoal generation within a single architecture. This is important because it overcomes key limitations of prior hierarchical methods by allowing latent subgoals to serve as intermediate reasoning steps while preserving final-goal awareness and enabling end-to-end optimization.
>
> > **W1. Architectural emphasis**
>
> Since CoGHP introduces MLP-Mixer into offline goal-conditioned RL, we considered it important to explain this choice. However, we agree that this may obscure the narrative flow around the core contribution of the chain-of-goals framework. In the final version, we will reduce repeated explanations and reorganize the presentation so that the main narrative focuses more clearly on CoGHP’s framework contribution.
>
> > **W2. Weak baselines**
>
> Following the reviewer’s suggestion, we evaluated OTA [5] and SAW [6] on challenging long-horizon environments. As shown below, CoGHP is strongest overall or highly competitive. OTA still relies on a single-subgoal abstraction, while SAW flattens hierarchy into a unified policy, which may be less suitable when explicit intermediate decomposition is important.
>
> | Env | OTA | SAW | CoGHP |
> | --- | --- | --- | --- |
> | pointmaze-giant-navigate-v0 | 72$\pm$6 | 68$\pm$8 | 79$\pm$8 |
> | antmaze-giant-navigate-v0  | 77$\pm$4 | 73$\pm$4 | 78$\pm$8 |
> | scene-play-v0 | 20$\pm$4 | 63$\pm$6 | 78$\pm$7 |
> | cube-triple-noisy-v0 | 2$\pm$1 | 17$\pm$3 | 42$\pm$3 |
>
> We agree that [1, 7, 8] are important recent works. However, they focus on inference-time planning or search, whereas CoGHP improves the policy architecture itself. We therefore view them as complementary rather than directly comparable baselines, since they target a different part of the overall problem. We will reflect this discussion in the `Related Work` section.
>
> > **W3. Multiple subgoals seem unnecessary**
>
> While H=1 performs best on the manipulation tasks, multiple subgoals still help in navigation. We attribute the drop in manipulation tasks to their greater demand for precise low-level control. Since CoGHP’s latent subgoals act not as explicit waypoints but as intermediate representations for action generation, overly fine subgoal decomposition can become counterproductive. We therefore view these results not as showing that subgoal sequences are unnecessary, but that the appropriate subgoal horizon is task-dependent.
>
> > **Q1. Training a separate low-level actor**
>
> We conducted an additional ablation in which CoGHP generates the latent subgoal embedding and a separate low-level actor predicts the primitive action. As shown below, performance drops substantially with a separate actor. This indicates that the gains of CoGHP do not arise solely from improved subgoal generation, but also from jointly learning subgoal prediction and action generation end-to-end within a unified model.
>
> | Env | CoGHP (seperate actor) | CoGHP |
> | --- | --- | --- |
> | antmaze-giant-navigate-v0 | 56$\pm$6 | 78$\pm$8 |
> | cube-double-noisy-v0 | 3$\pm$2 | 54$\pm$5 |
>
> > **Q2. Additional results on visual tasks**
> >
>
> We additionally evaluated CoGHP on visual-antmaze-large and visual-cube-double. CoGHP is comparable to HIQL on the former and slightly better on the latter, showing that it remains competitive in long-horizon visual settings. In the final version, we will discuss these results together with the point that combining CoGHP with stronger visual encoders could allow its hierarchical reasoning to be more effectively leveraged in visual environments.
>
> | Env | HIQL | CoGHP |
> | --- | --- | --- |
> | visual-antmaze-large-navigate-v0 | 53$\pm$9 | 53$\pm$3 |
> | visual-cube-double-noisy-v0 | 59$\pm$3 | 62$\pm$3 |
>
> > **Q3. Results on the stitch and explore datasets**
>
> CoGHP outperforms HIQL on the explore datasets but is less competitive on the stitch datasets. In contrast, CompDiffuser is specifically designed to stitch short trajectory chunks at test time and therefore performs strongly in these settings. However, this comes at the cost of substantial test-time computation, whereas CoGHP supports real-time execution. This suggests that combining these two approaches would be an interesting direction for future work.
>
> | Env | HIQL | CompDiffuser | CoGHP |
> | --- | --- | --- | --- |
> | antmaze-medium-explore-v0 | 37$\pm$10 | 81$\pm$2 | 58$\pm$3 |
> | antmaze-large-explore-v0 | 4$\pm$5 | 27$\pm$1 | 17$\pm$4 |
> | antmaze-medium-stitch-v0 | 94$\pm$1 | 96$\pm$2 | 91$\pm$1 |
> | antmaze-large-stitch-v0 | 67$\pm$5 | 86$\pm$2 | 60$\pm$6 |

---

> > ### Author Rebuttal · Reviewer_Dakr · 2026-04-04
> >
> > While I appreciate the additional results on stitch and visual environments, I still have significant concerns about the CoT paradigm and the claims in the paper. As the authors themselves stated, their method is an "improvement to the policy architecture" and not really chain-of-thought as is commonly defined (as also noted by Reviewer Gj9N). Even though it might be loosely inspired by the general concept of problem decomposition, it is quite confusing to call it CoT.
> >
> > My most significant concern is about the lack of performance gains from multiple subgoals (W3), especially in compositional manipulation where one would expect it to help the most, given the structure of multiple atomic tasks composed in sequence. This is the central motivating premise of the paper but the paper itself does not support these conclusions. The authors' rebuttal that it demonstrates the importance of selecting a good subgoal horizon and not that multiple subgoals are unnecessary seems to be directly contradicted by the joint sensitivity analysis of k and H in Table 5 of the Appendix, where cube-double-noisy and scene are universally worse for H > 1, for all values of k, and quite significantly as well.
> >
> > Additionally, as the authors showed in the response to Q1, many of the performance gains can be attributed to better one-step subgoal-conditioned action generation (akin to a better low actor) from parameterizing the subgoal generator and the primitive action policy with the same network. I appreciate this finding and think it is a good contribution that would be of interest to the community when paired with insights on why this occurs. However, these results do not match the narrative and positioning of the paper as currently written (especially the first Introduction paragraph, e.g., line 21 onwards on the RHS and the analysis in line 366, LHS), so I unfortunately cannot raise my score.

---

> > > ### Author Response · Authors · 2026-04-06
> > >
> > > We appreciate the reviewer’s additional comments. Below, we address the main issues raised by the reviewer.
> > >
> > > ---
> > >
> > > > **Confusion around the CoT terminology**
> > >
> > > We understand the reviewer’s concern about the CoT terminology. However, the paper does not claim that CoGHP applies CoT as it is commonly defined in NLP. Rather, **CoGHP is a framework that brings the structural ideas of CoT, specifically stepwise decomposition and sequential conditioning, into hierarchical offline goal-conditioned RL.**
> > >
> > > Concretely, given a state and a final goal, CoGHP autoregressively generates a sequence of latent subgoals, where each subgoal serves as intermediate conditioning for subsequent predictions. CoGHP is therefore better described as a CoT-inspired hierarchical framework. This also matches the paper’s original phrasing that we “draw inspiration from the chain-of-thought paradigm.”
> > >
> > > Many recent works in vision-language-action have also explored applying the ideas of CoT to embodied control [1, 2, 3]. In particular, Embodied Chain of Thought (ECoT) [1] reinterprets these ideas for embodied control and performs action prediction by autoregressively predicting embodied intermediate representations such as subtasks, movements, and gripper positions. While CoGHP is aligned with this line of work, its contribution lies in extending this CoT-inspired approach to the offline goal-conditioned RL setting.
> > >
> > > However, we agree that some parts of the paper may make CoGHP read as if it directly applies CoT from NLP. In the final version, we will revise the relevant wording to make it clearer that CoGHP is a framework inspired by CoT.
> > >
> > > ---
> > >
> > > > **Interpretation of the multiple-subgoal results**
> > >
> > > As the reviewer notes, multiple subgoals do not necessarily lead to performance gains in the manipulation tasks. That said, the strength of CoGHP is not limited to generating multiple subgoals. **CoGHP’s more fundamental contribution lies in its unified autoregressive structure, which handles multiple intermediate decisions within a single framework while preserving final-goal awareness throughout the process.** This is further supported by the fact that CoGHP outperforms the baselines on manipulation tasks even with a single-subgoal setting (H = 1).
> > >
> > > A key point from `Sections C.2 and C.3`, which present ablations on the number of subgoals and subgoal step, is that the appropriate subgoal setting depends on the task. While H = 1 gives the best performance in the manipulation tasks, the navigation tasks clearly show that multiple subgoals can contribute to performance gains. We explained this difference by noting that excessive hierarchical decomposition can hinder precise control in manipulation tasks. Therefore, our interpretation of these results was not that multiple subgoals are always beneficial, nor that they are unnecessary, but that the appropriate subgoal horizon is task-dependent.
> > >
> > > Here, the term subgoal horizon, as used in `Section C.3`, refers to both the number of generated subgoals H and the subgoal step k. Under this definition, `Table 5` is consistent with our rebuttal that the appropriate subgoal horizon depends on the task, as both H and k affect performance with H having a stronger effect. We discussed this point in the `Limitations section`, together with related directions for future work.
> > >
> > > ---
> > >
> > > > **Implications of the separate-actor ablation**
> > >
> > > We appreciate the reviewer’s recognition that the result from the separate-actor ablation may be of interest to the community. The substantial performance drop with a separate actor suggests that the strength of CoGHP is not explained simply by better latent subgoal generation or a better low-level actor. **This instead highlights the importance of the unified autoregressive structure proposed in CoGHP, which we view as a key aspect that distinguishes CoGHP from prior hierarchical RL approaches.**
> > >
> > > ---
> > >
> > > Thank you again for the thoughtful comments. This feedback has helped us clarify our contributions more explicitly. We hope that our response helps address the reviewer’s concerns.
> > >
> > > **References**
> > >
> > > [1] Zawalski, Michał, et al. "Robotic control via embodied chain of thought reasoning." *arXiv preprint arXiv:2407.08693* (2024).
> > >
> > > [2] Chen, William, et al. "Training strategies for efficient embodied reasoning." *arXiv preprint arXiv:2505.08243* (2025).
> > >
> > > [3] Team, Gemini Robotics, et al. "Gemini robotics: Bringing ai into the physical world." *arXiv preprint arXiv:2503.20020* (2025).

---

### Decision · Program_Chairs · 2026-04-30

**Decision:**

Accept (regular)

**Comment:**

The paper presents a novel approach to hierarchical decomposition of offline goal-conditioned reinforcement learning problems. The approach is to train a module (MLP-Mixer) that predicts the next subgoal to achieve given the current state, final goal and sequence of previously generated subgoals. The subgoal sequence is then generated backwards in a regressive manner starting with the final goal. Another novelty is to use a single neural network architecture for all hierarchy levels. The resulting algorithm CoGHP is tested in a series of experiments on GCRL benchmarks from the OGBench suite.

The reviewers agree that the autoregressive approach to subgoal generation is interesting, that the choice of architecture is well-motivated and that the experimental results are strong. The paper provides additional evidence that subgoal decomposition can significantly increase learning performance in long-horizon tasks. The perceived weaknesses are that multiple subgoals are not needed in most domains, and that the approach could have been tested in more domains with long horizons to further validate the effect of subgoal sequence generation. My impression is mostly positive and I think the paper makes a solid and original contribution to the field, so my recommendation is to accept the paper for publication at ICML.